# Refining the adipose progenitor cell landscape in healthy and obese visceral adipose tissue using single-cell gene expression profiling

Dong Seong Cho, Bolim Lee, Jason D Doles

Obesity is a serious health concern and is associated with a reduced quality of life and a number of chronic diseases, including diabetes, heart disease, stroke, and cancer. With obesity rates on the rise worldwide, adipose tissue biology has become a top biomedical research priority. Despite steady growth in obesity-related research, more investigation into the basic biology of adipose tissue is needed to drive innovative solutions aiming to curtail the obesity epidemic. Adipose progenitor cells (APCs) play a central role in adipose tissue homeostasis and coordinate adipose tissue expansion and remodeling. Although APCs are well studied, defining and characterizing APC subsets remains ambiguous because of ill-defined cellular heterogeneity within this cellular compartment. In this study, we used single-cell RNA sequencing to create a cellular atlas of APC heterogeneity in mouse visceral adipose tissue. Our analysis identified two distinct populations of adipose tissue–derived stem cells (ASCs) and three distinct populations of preadipocytes (PAs). We identified novel cell surface markers that, when used in combination with traditional ASC and preadipocyte markers, could discriminate between these APC subpopulations by flow cytometry. Prospective isolation and molecular characterization of these APC subpopulations confirmed single-cell RNA sequencing gene expression signatures, and ex vivo culture revealed differential expansion/differentiation capabilities. Obese visceral adipose tissue featured relative expansion of less mature ASC and PA subpopulations, and expression analyses revealed major obesity-associated signaling alterations within each APC subpopulation. Taken together, our study highlights cellular and transcriptional heterogeneity within the APC pool, provides new tools to prospectively isolate and study these novel subpopulations, and underscores the importance of considering APC diversity when studying the etiology of obesity.

## Introduction

Mammalian adipose tissue is generally divided into two types: white adipose tissue (WAT) and brown adipose tissue. Substantial heterogeneity exists within these two general subtypes; WAT, for example, can be subdivided into subcutaneous (SWAT) and visceral (VWAT) depots, and cells within these depots can vary depending on precise anatomical locations. WAT is capable of remarkable expansion, a property that left unchecked results in excess adipose tissue accumulation, obesity, and related pathologies. The two main forces that underlie WAT expansion are adipocyte hyperplasia and adipocyte hypertrophy. The latter involves increases in adipocyte size/volume, largely fueled by shifts in the balance between lipid storage (lipogenesis) and lipid breakdown (lipolysis). In contrast, adipocyte hyperplasia involves an increase in adipocyte number, a result of aberrant adipose progenitor cell (APC) expansion, differentiation, and self-renewal programs. Indeed, recent work suggests that hyperplasia, as opposed to hypertrophy, is the major contributor to expansion of VWAT in human obesity (Spalding et al, 2008; Arner et al, 2013). Thus, understanding fundamental APC properties is highly relevant as obesity-related research moves forward.

The process of APC differentiation has been extensively studied in vitro using both immortalized cell systems such as 3T3-L1 and 3T3-F442A, as well as primary cell culture systems that typically rely on flow cytometry–based isolation of enriched APCs from various mammalian adipose depots. In either system, immature progenitor cells proceed along a well-defined maturation trajectory, starting with proliferation/expansion from highly proliferative and multipotent adipose-derived stem cells (ASCs), proceeding to cell cycle arrest and early differentiation giving rise to lineage-committed progenitors, termed "preadipocytes" (PAs), and culminating in terminal differentiation as a mature adipocyte. Although this differentiation process is fairly well characterized, there remains a substantial degree of variability when it comes to APC differentiation capacity/potential. For example, numerous studies point to replication and differentiation differences in cultured preadipocytes/APCs isolated from SWAT versus VWAT (Tchkonia et al, 2002, 2006; Baglioni et al, 2009; Toyoda et al, 2009). Differences remain even when comparing different VWAT depots such as omental and mesenteric depots (Tchkonia et al, 2005, 2007; Palmer & Kirkland, 2016). Why are such differences observed? Some variation may be explained by environmental (depot-specific) microenvironment influences that persist upon limited in vitro culture. Others propose that APC subpopulations

Department of Biochemistry and Molecular Biology, Mayo Clinic, Rochester, MN, USA

Correspondence: Doles.Jason@mayo.edu

with differing proliferation/differentiation potential exist and are differentially abundant in a depot-specific manner (Tchkonia et al, 2005; Rodeheffer et al, 2008; Boumelhem et al, 2017). With respect to the latter hypothesis, rigorous analyses that assess the extent of APC heterogeneity are needed to more accurately interpret functional differences in depot-specific APC potential.

Bulk transcriptional profiling of APCs has revealed extensive transcriptional variability in APCs isolated from different adipose depots (Macotela et al, 2012; Hepler et al, 2017) or in pathological settings (Patel et al, 2016; Hepler et al, 2017). Few studies, however, have examined APC heterogeneity on a global transcriptional level. Single-cell RNA sequencing (scRNA-seq) is rapidly gaining traction as a methodology that can be leveraged to query tissue or cell type heterogeneity on a global level. Indeed, scRNA-seq has been used to investigate induced pluripotent stem cell heterogeneity (Nguyen et al, 2018), as well as progenitor cell diversity in multiple adult tissue types (Wen & Tang, 2016), including adipose tissue (Burl et al, 2018; Hepler et al, 2018; Schwalie et al, 2018; Merrick et al, 2019). Although recent scRNA-seq studies have significantly advanced our understanding of cellular diversity (APCs, immune cells, fibro-inflammatory cells, mesothelial-like cells, and adipogenesis-regulatory cells) within adipose tissue (Burl et al, 2018; Hepler et al, 2018; Schwalie et al, 2018; Merrick et al, 2019), the extent of molecular and functional heterogeneity within the APC compartment is unclear. In the present study, we use scRNA-seq to study the transcriptional diversity of more than 4,500 freshly isolated APCs from normal and obese visceral adipose tissue. Leveraging the unbiased nature of sequencing-based transcriptomics, we identify, validate, and characterize novel APC subpopulations. We compare these novel APC subpopulations with other APC subpopulations recently identified in four independent scRNA-seq studies (Burl et al, 2018; Hepler et al, 2018; Schwalie et al, 2018; Merrick et al, 2019). We further demonstrate substantial subpopulation rearrangement in the context of obesity. We propose that these rearrangements may underlie functional differences between normal and obese APCs and may contribute to the etiology of obesity.

## Results and Discussion

### Single-cell analyses of APCs isolated from normal and obese murine adipose tissue

In this study, we modeled diet-induced obesity (DIO) using a short-term, high-fat dietary formulation that featured elevated fat (40–45% kcal) and sucrose (36.8% by weight) content. After 13 d in control or DIO conditions, DIO mice gained 17.6% ± 2.6% of their total body weight, a significant increase over that observed in control diet–fed mice (2.6% ± 3.6%) (Fig 1A). Epididymal fat pad wet weight was increased ~2-fold in DIO mice compared with controls (Fig 1B). In preparation for single-cell sequencing studies, we queried adipose tissue progenitor cell (APC) abundance in adipose tissue from control versus DIO mice. APCs (Sca1$^{pos}$/CD31$^{neg}$/CD45$^{neg}$/Ter119$^{neg}$) were 32.9% and 20.3% of viable single cells in the adipose tissue from control and DIO mice, respectively (Fig 1C and D). Although the proportion of APCs was lower in the DIO mice, the absolute number of APCs in total tissue was similar in both conditions (Fig 1D). These

results show that whereas total tissue mass increases during obesity, the size of the progenitor cell pool remains constant.

To query the extent of APC heterogeneity in normal and obese visceral adipose tissue, we used single-cell, droplet-based RNA sequencing (scRNA-seq) (Zheng et al, 2017). Sca1$^{pos}$/CD31$^{neg}$/CD45$^{neg}$/Ter119$^{neg}$ APCs were isolated from epididymal fat pads collected form control diet and DIO mice (Fig S1A). We successfully captured and profiled 2,636 APCs (control) and 2,143 APCs (obese) with >700 million reads per sample (Fig S1B). More than 280,000 reads per cell were generated for both samples, numbers that exceeded previously reported values (~100,000 reads) required to obtain the maximum median number of genes detected per cell (Zheng et al, 2017), thus permitting robust downstream differential transcript expression analyses. We then performed a principal component analysis (PCA), followed by a t-distributed stochastic neighbor embedding (tSNE) projection (Macosko et al, 2015; Yan et al, 2017; Zheng et al, 2017) on all 4,779 cells to create a comprehensive atlas of normal and obese APCs (Figs 1E and S2). After the tSNE projection, the cells from each condition (control and DIO) were shown separately against the backdrop of the aggregated dataset. We applied k-means clustering on the first 10 principal components computed from the PCA, a widely used method to cluster scRNA-seq datasets (Grun et al, 2015; Zurauskiene & Yau, 2016; Kiselev et al, 2017), as the first step in identifying potential APC subpopulations (Figs 1E and S2). Average silhouette width analysis to determine the optimal number of clusters in k-means clustering (Rousseeuw, 1987) was performed and indicated that k = 2 was the most appropriate cluster number (Fig S2B). Importantly, however, k = 2 failed to separate a cell cluster expressing low levels of Ly6a (Sca1 transcript) and multiple housekeeping genes that were clustered in a clearly separate area within tSNE projections (see arrows in Fig S2C) (Fig S2A and C). We determined that these cells could be separated using k = 6 or greater (Figs 1E and S2A). At k = 7, however, one cluster contained only two cells in DIO tissue and zero cells in control tissue (see blue dots in Fig S2A)—a number too low for statistical analysis to find differentially expressed genes and to determine population identity. Thus, we chose k = 6 for subsequent analyses to define potential subclusters within these two main subpopulations and to determine the extent to which these subclusters exhibited transcriptomic or functional differences.

TSNE projections revealed significant subpopulation shifts in DIO versus control APCs (Fig 1E and F). In DIO APCs, the proportions of clusters 1, 2, and 4 were decreased from 34%, 22%, and 16% (control) to 21%, 12%, and 8%, respectively. In contrast, the proportions of clusters 3, 5, and 6 were increased from 1%, 19%, and 8% to 3%, 28%, and 28%, respectively. These data show that APCs are transcriptionally diverse that they can be bioinformatically clustered into six groups and that DIO alters the relative proportions of these APC subpopulations.

### Expression analysis broadly categorized APC subpopulations into two clusters of adipose-derived stem cells and preadipocytes

We next queried subpopulation similarity by performing hierarchical clustering based on the average expression levels of cells in each cluster from the control condition (Fig 2A). We chose to focus this analysis on control APCs as we did not want the confounding

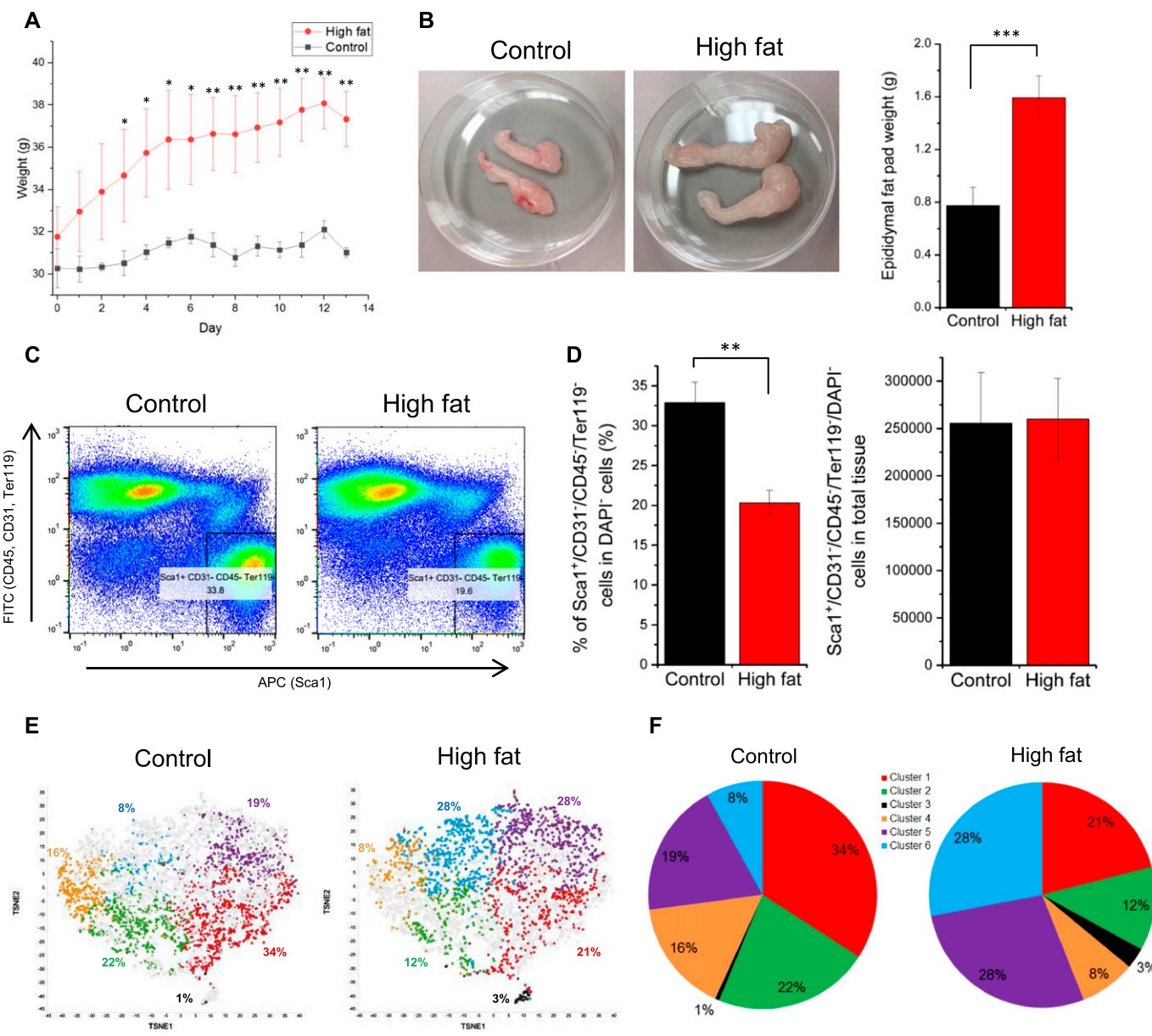

**Figure 1. scRNA-seq analysis of APCs from control and obese mice.**
**(A)** Body weight of mice on control and high-fat/DIO diets (n = 4; error bars: SD). Statistical tests were performed by *t* test assuming unequal variance. *$P$-value < 0.05; **$P$-value < 0.01. **(B)** Epididymal fat pads isolated from control and high-fat/DIO mice (n = 4; error bars: SD; *t* test assuming unequal variance; ***$P$-value < 0.001). **(C)** Flow cytometry analysis of cells dissociated from epididymal fat pads from control and high-fat/DIO mice and gating strategy to isolate APC populations (Sca1[pos]/CD31[neg]/CD45[neg]/Ter119[neg] cells). **(D)** Quantification of Sca1[pos]/CD31[neg]/CD45[neg]/Ter119[neg] cells in DAPI[neg] cells (left) and Sca1[pos]/CD31[neg]/CD45[neg]/Ter119[neg] cells in total tissue (right) (n = 3; error bars: SD; *t* test assuming unequal variance; **$P$-value < 0.01). **(E)** tSNE projections of total 4,779 sequenced APCs from control (left) and high-fat/DIO mice (right). **(F)** Pie charts of six clusters identified from APCs in control (left) and high-fat/DIO mice (right) by tSNE projection.

effect of DIO to skew these initial subpopulation similarity comparisons. APC subpopulation shifts as a consequence of DIO are addressed later in this study. Hierarchical clustering analysis showed that cluster 3, which was the least abundant in both control and DIO samples, was significantly distinct from the rest of the clusters, whereas the rest of the APCs formed two main groups (clusters 2/4/6 and clusters 1/5). Pairwise comparison of these clusters corroborated the hierarchical clustering results (Fig 2B). Nonnegative matrix factorization (NMF), which clusters cells into a

defined number of groups (Brunet et al, 2004) showed that the cophenetic correlation coefficient, a coefficient representing robustness of clustering, was the highest at two groups, indicating that two groups optimally clusters the overall APC pool (Fig S3).

We next used differential gene expression analyses to characterize these six clusters. First, we compared cluster 3 versus all other clusters because cluster 3 was the rarest and most distantly clustered subpopulation. Differentially expressed genes were determined by significance analysis of microarrays (SAM) (Tusher et al, 2001)

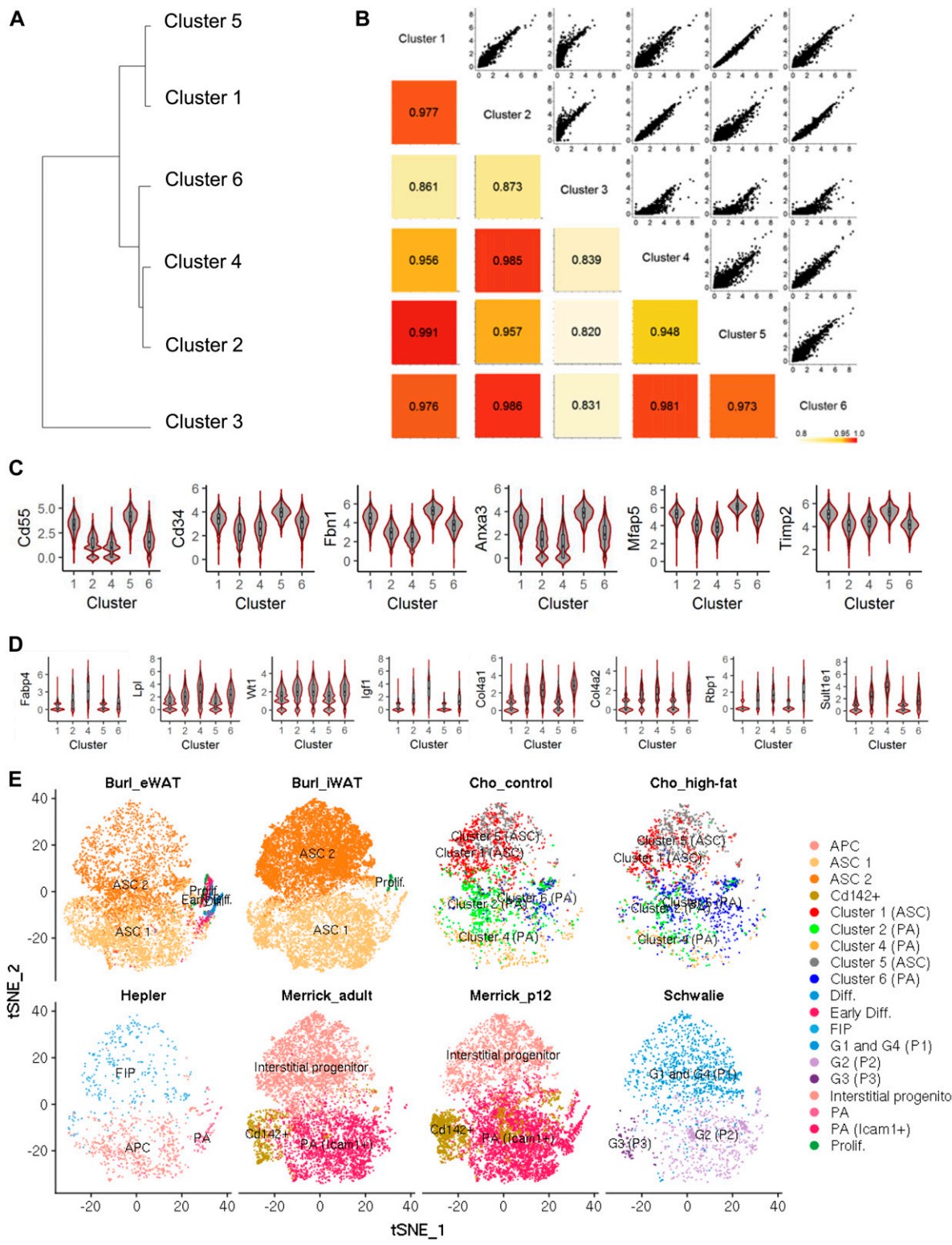

**Figure 2. APC subpopulation analysis.**
**(A)** Hierarchical clustering of each subpopulation found from tSNE analyses. **(B)** Pairwise comparison of each cluster. Pearson correlation coefficients for each pair are shown with colors to represent similarity of two clusters in each pair. **(C, D)** Expression levels of highly expressed genes in cluster 1 and 5 (C), and in cluster 2, 4, and 6 (D) that are associated with adipose derived stem cells, preadipocytes, adipocytes, or adipose tissue. Expression levels are shown with log2-normalized unique molecular identifier counts in this figure and in the rest of following figures. **(E)** Comparative analysis with four publicly available scRNA-seq datasets visualized on tSNE projections.

with threshold of fold change >1.5 and q-value = 0. SAM identified several up-regulated transcripts in cluster 3, including Hba-a1, Hba-a2, Hbb-bs, and Hbb-bt that were highly expressed in 2/16 cells in cluster 3 (Fig S4A and B). All of these transcripts encode hemoglobin proteins, suggesting that cluster 3 may include erythroid lineage cells. Furthermore, cluster 3 had low expression of Ly6a (Sca1 transcript) (Fig S4C) and expression levels of several widely used house-keeping genes were low only in cluster 3 (Fig S4D). These results indicate that cluster 3 consists of cells which are not bona fide APCs (Sca1$^{pos}$/CD31$^{neg}$/CD45$^{neg}$/Ter119$^{neg}$ cells) but rather outliers because of either poor cDNA library amplification or cellular contamination during cell isolation. Therefore, cluster 3 was excluded from subsequent analyses.

We then compared the two main APC groups: clusters 1/5 versus clusters 2/4/6. SAM identified 145 up-regulated genes and 80 down-regulated genes in clusters 1/5 (see the full list of these genes in Table S1). Among these genes, clusters 1/5 exhibited high expression of Cd55, a human adipose-derived stem cell (ASC) marker, and Cd34, a human and mouse ASC marker (Figs 2C and S5, and Table S11). In addition, clusters 1/5 exhibited elevated transcript levels of Fbn1, Anxa3, Mfap5, and Timp2, all of which are reported to be down-regulated upon adipocyte differentiation (Table S11). Adipose tissue-resident interstitial progenitor markers, Dpp4 and Pi16 (Merrick et al, 2019), were also highly expressed in clusters 1/5 (Fig S5A). In contrast, clusters 2/4/6 exhibited elevated expression of the mature adipocyte markers, Fabp4 and Lpl, and the pre-adipocyte markers, Wt1 and Rbp1 (Figs 2D and S5B, and Table S11). In addition, expression levels of the following adipocyte-associated transcripts were elevated in clusters 2/4/6: Igf1, which is involved in adipocyte homeostasis, Col4a1 and Col4a2, both of which are up-regulated during adipocyte differentiation, and Sult1e1 which promotes adipocyte differentiation (Table S11). Altogether, these expression patterns indicate that clusters 1/5 are likely more primitive, undifferentiated ASCs, whereas clusters 2/4/6 are more committed preadipocytes.

Next, we combined our dataset with four published adipose tissue/APC scRNA-seq datasets (Burl et al, 2018; Hepler et al, 2018; Schwalie et al, 2018; Merrick et al, 2019) to contextualize our newly defined clusters against the backdrop of previously published data (Fig 2E). This comparative analysis showed that clusters 1/5 (ASC) are similar to "ASC 2" in Burl et al (2018), "fibro-inflammatory progenitor" (FIP) in (Hepler et al, 2018), "interstitial progenitor" in Merrick et al (2019), and "G1 and G4" ("P1") in Schwalie et al (2018), whereas clusters 2/4/6 (PA) are similar to "ASC 1" in Burl et al (2018), "APC" and "PA" in (Hepler et al, 2018), "Icam1$^{+}$ PA" in Merrick et al (2019), and "G2" ("P2") in Schwalie et al (2018). In comparison with ASC 1, ASC 2 was shown to contain fewer cells expressing adipogenic transcription factors, Cebpa and Pparg, as well as cells with lower expression of ECM genes, including Col4a1 and Bgn (Burl et al, 2018). FIP was reported to have limited adipogenic capacity (Hepler et al, 2018), whereas interstitial progenitors, G1 and G4 (P1) populations had more stem cell–specific or immature properties than other populations in either study (Schwalie et al, 2018; Merrick et al, 2019). Altogether, the ASC (clusters 1/5) and PA (clusters 2/4/6) populations in our study closely resemble other APC subpopulations identified in previous scRNA-seq studies, thus corroborating our broad, higher order (ASC versus PA) classification scheme.

Next, we performed differential expression (DE) analysis using clusters 1 and 5 to assess heterogeneity between these ASC populations (Fig 3A and Table S2). Cluster 5 had elevated transcript levels of ASC markers Cd34 and Cd55, as well as ASC transcripts associated with negative regulation of adipocyte differentiation, such as Sparc, Cyr61, and Klf2 (Figs 3A and S5C, and Table S11). Furthermore, in cluster 5, we observed higher expression of transcripts normally down-regulated during adipocyte differentiation (Fstl1, Fbn1, Cfl1, and Anxa3) and lower expression of transcripts up-regulated during adipocyte differentiation (Mt1 and Mt2) (Figs 3A and S5C, and Table S11). Finally, pathway analysis of DE transcripts associated with cluster 5 revealed processes associated with negative regulation of adipocyte differentiation, such as Rho signaling and ILK signaling (Fig S6A and Table S11). Together, these results indicate that ASCs can be subdivided into two subpopulations, primarily based on relative adipogenic maturity.

Similar analyses were performed using PA clusters 2, 4, and 6 (Tables S3–S5 and Figs 3B and C and S5D). Cluster 6 exhibited higher expression of Meg3, a long noncoding RNA that inhibits human adipocyte differentiation, as well as Mfap5, Fbn1, Thy1, Fstl1, and Col6a2—transcripts that are typically down-regulated during adipocyte differentiation (Table S11). Pathways linked to negative regulation of adipocyte differentiation were associated with cluster 6 DE genes (Fig S6B). These results suggest that like ASC cluster 5, PA cluster 6 represents a relatively less mature APC subpopulation. In contrast, cluster 4 exhibited elevated expression of the mature adipocyte markers, Apoe and Fabp4, as well as transcripts involved in mature adipocyte function, such as Igf1, Jun, and Fos (Table S11). Furthermore, we observed high cluster 4 expression of Sult1e1, a positive regulator of adipocyte differentiation, and Trf, a transcript known to increase during adipocyte differentiation (Table S11). Pathway analysis of cluster 4 DE genes identified Aryl signaling and liver X receptor/retinoid X receptor associated signaling–two pathways linked to mature adipocyte function (Fig S6C and Table S11). These expression data, therefore, suggest that cluster 4 represents the most mature PA subpopulation. Overall, our single-cell analyses of APCs identified distinct ASC and PA subpopulations, each containing subclusters exhibiting differing maturation profiles. Based on the distribution of each cluster in control and DIO conditions (Fig 1F), our analysis indicates the following: (1) the relative proportions of ASCs (cluster 1 and 5) and PAs (cluster 2, 4, and 6) are similar between control and DIO conditions, and (2) DIO results in an increase in the proportion of less mature subpopulations in both ASC (cluster 5) and PA (cluster 6) populations, whereas more mature subpopulations (clusters 1, 2, and 4) are under-represented.

### Prospective isolation, validation, and functional assessment of novel APC subpopulations

We next sought to identify and potentially use cell surface makers capable of discriminating between these newly identified APC subpopulations. Based on the DE genes associated with each cluster (Tables S1–S5), cell-surface associated cluster of differentiation transcripts Cd55, Cd81, and Cd9 were predicted to be sufficient to distinguish these five clusters as follows: cluster 1 (Cd55$^{high}$/Cd81$^{low}$), cluster 5 (Cd55$^{high}$/Cd81$^{high}$), cluster 2 (Cd55$^{low}$/Cd9$^{low-mid}$/Cd81$^{low}$), cluster 4 (Cd55$^{low}$/Cd9$^{low-mid}$/Cd81$^{high}$), and cluster 6 (Cd55$^{low}$/Cd9$^{high}$) (Fig 4A).

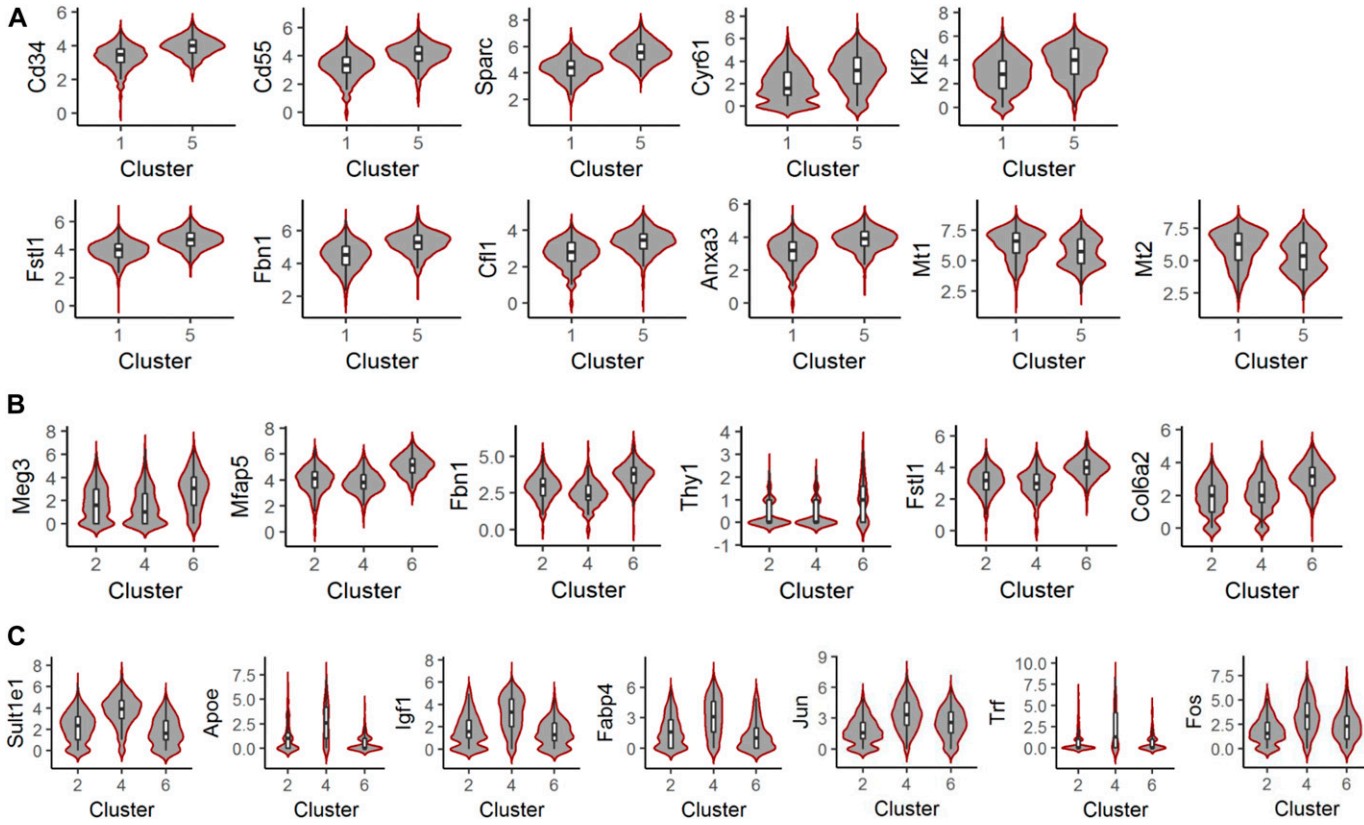

**Figure 3. Differentially expressed genes between APC subpopulations.**
**(A, B, C)** Expression levels of differentially expressed genes in cluster 5 versus cluster 1 (A), in cluster 6 versus cluster 2 and 4 (B), and in cluster 4 versus cluster 2 and 6 (C).

We, therefore, labeled and further subdivided APCs (Sca1[pos]/CD31[neg]/CD45[neg]/Ter119[neg]) using antibodies against CD55, CD81, and CD9. Flow cytometry analysis confirmed that APCs separated into five distinct populations based on these new APC markers (Fig 4B).

Prospective isolation of these five subpopulations was performed and expression of DE genes found in the above cluster transcriptome analysis was queried by quantitative RT-PCR (Fig 4C–F). Consistent with the transcriptome analysis, most of the markers categorizing the five clusters into ASC (Fig 2C) and preadipocyte populations (Fig 2D) were elevated in clusters 1 and 5 (Fig 4C and D), and in clusters 2, 4, and 6 (Fig 4E and F), respectively. Moreover, we validated several other top DE genes in the comparison of clusters 1 and 5 versus clusters 2, 4, and 6 (Table S1 and Fig 4D and F).

Flow-sorted APC subpopulations were then plated into 96-well plates to directly compare ex vivo expansion and differentiation capabilities (Fig 5). ASCs (clusters 1 and 5) and the immature PA subpopulation (cluster 6) exhibited significantly faster proliferation rates than other two, more mature PA subpopulations (clusters 2 and 4) (Fig 5A). Cluster 4, the most mature preadipocyte subpopulation, expanded significantly more slowly than all other clusters. In addition, upon induction of adipocyte differentiation, the two mature PA subpopulations (clusters 2 and 4) had significantly greater differentiation capacities than ASCs (clusters 1 and 5) (Fig 5B–D and Videos 1–5). Hence, our data show that both ex vivo expansion rates and differentiation capacities correlate with

transcriptomic adipogenic "maturity," whereby proliferative capacity declines, whereas differentiation capacity increases as APCs progress toward a more mature state. It remains unclear, however, whether these five clusters represent distinct states on a trajectory of differentiation of ASCs toward mature adipocytes or whether they represent independent subpopulations resident in adipose tissue. In addition to more rigorously assessing adipogenic differentiation capacity between ASC and PA subpopulations, it will be important to demonstrate multipotent differentiation capacities of ASCs/PAs toward non-adipogenic lineages, including osteogenic and chondrogenic lineages, as ASCs/PAs typically possess these multipotent differentiation capacities (Cawthorn et al, 2012).

Recent scRNA-seq analyses suggest that APCs comprise multiple distinct subpopulations. First, Burl et al (2018) described two main APC subpopulations (ASC 1 and ASC 2) with DE of genes associated with ECM production and proteolysis (Burl et al, 2018). This is similar to our finding that ASCs and PAs exhibited DE transcripts involved in ECM production. In a second study, scRNA-seq analysis on Pdgfrb-expressing cells in visceral adipose tissue revealed a subset of APCs that resemble committed PAs (Hepler et al, 2018). Third, a few key markers distinguishing each subpopulation (Cd55 and Cd34 for ASCs, and Fabp4 for PAs) in our study overlap with markers reported in Schwalie et al (2018) that defined two of their three mouse stromal cell subpopulations (Cd55[high]/Cd34[high] G1 and G4, and Fabp4[high] G2) identified from their scRNA-seq data (Schwalie et al,

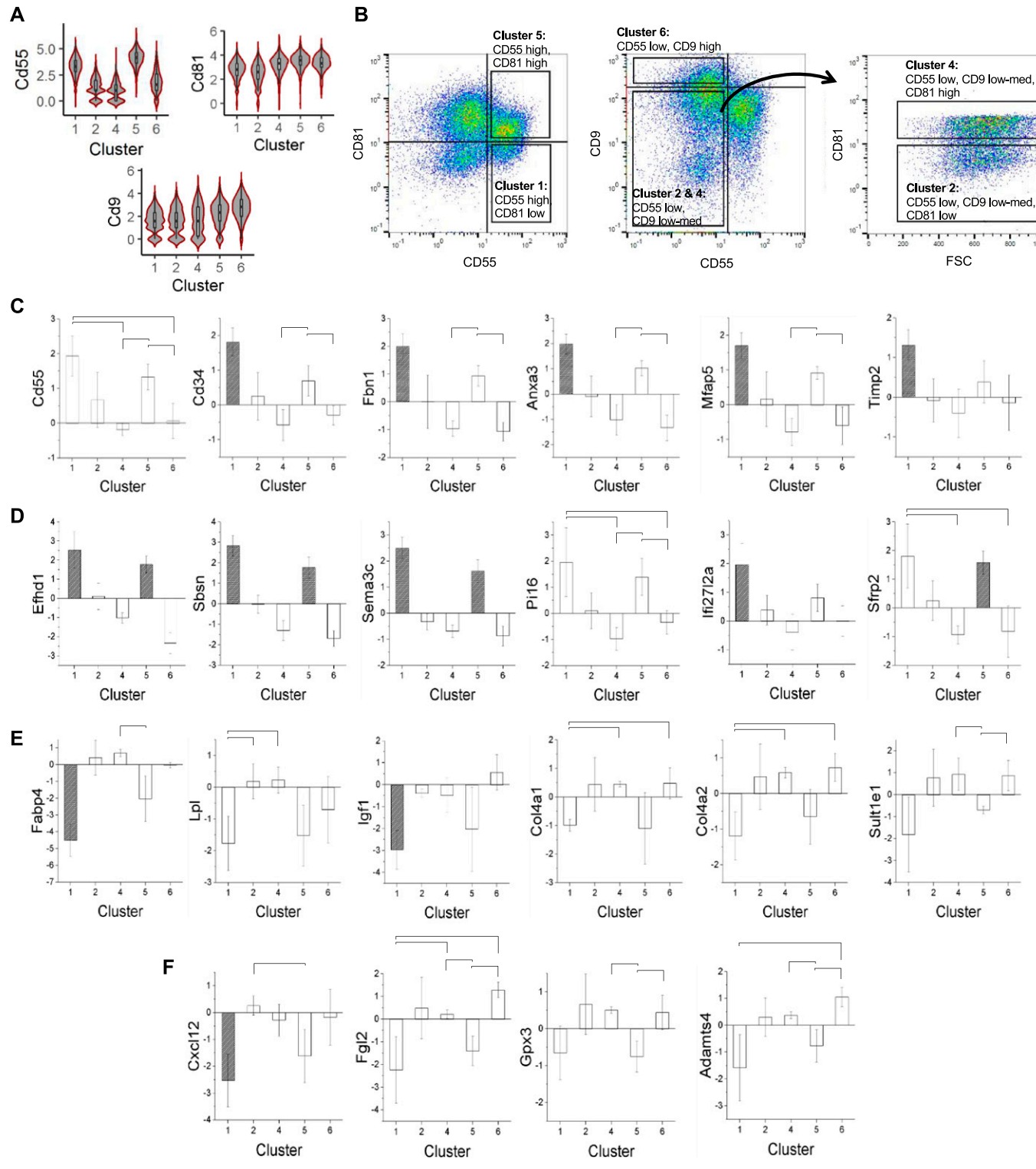

**Figure 4. Prospective isolation and validation of APC subpopulations.**
**(A)** Surface marker candidates that potentially distinguish each cluster of APCs. **(B)** Flow cytometry analysis of CD55, CD81, and CD9 expression in APCs (Sca1$^{pos}$/CD45$^{neg}$/CD31$^{neg}$/Ter119$^{neg}$) and gating strategy to isolate each subpopulation. **(C, D, E, F)** Validation of adipose stem cell (C, D) and preadipocyte (E, F) markers identified from scRNA-seq analysis (n = 3). **(C, E)**: Genes known to be associated with adipose stem cells, preadipocytes, or adipose tissues. **(D, F)**: Novel potential markers for adipose stem cells and preadipocytes. Transcript levels are quantified as log$_2$ expression level of each gene relative to a housekeeping gene, Tuba1b, and data are shown as relative expression compared with presorted APCs (Sca1$^{pos}$/CD45$^{neg}$/CD31$^{neg}$/Ter119$^{neg}$). Colored column of cluster 1 or 5 shows significantly different expression (n = 3; P-value < 0.05, t test assuming equal variance) in all pairwise comparison with cluster 2, 4, and 6. Marked comparison indicates significantly different comparison.

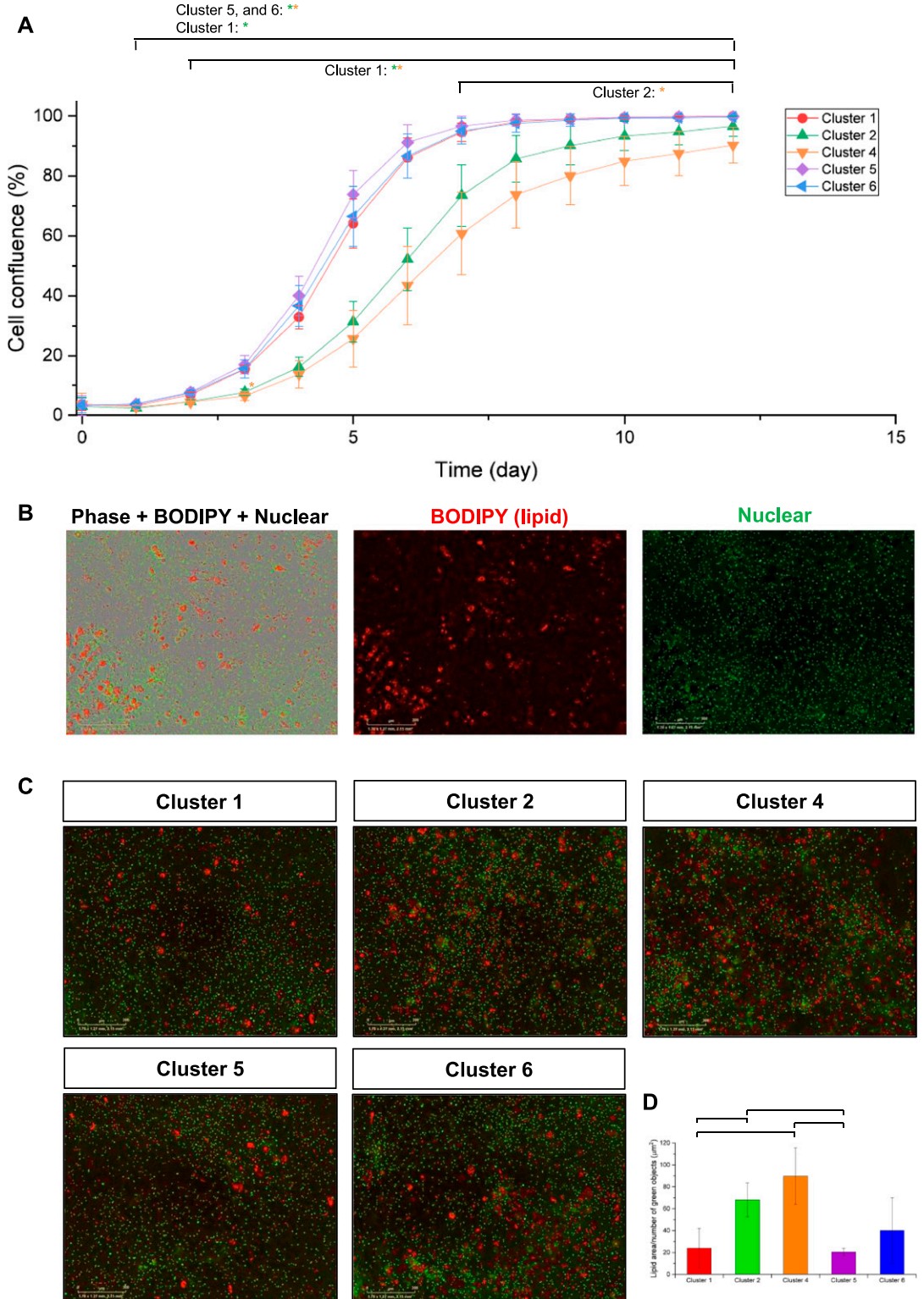

**Figure 5. APC subpopulations have distinct proliferation and differentiation potential.**
**(A)** A graph quantifying the proliferative potential of each cluster. Marked comparison indicates significantly different comparison ($P$-value < 0.05, $t$ test assuming equal variance). **(B)** Representative images of lipid and nuclear staining after differentiation of APC subpopulations. Scale bars: 300 $\mu$m. **(C, D)** Quantification of differentiation capacities of each APC subpopulation. **(C)** Representative images for lipid and nuclear staining after 10 d of differentiation. Red = BODIPY, green = nuclei. Scale bars: 300 $\mu$m. **(D)** Comparison of lipid area normalized with number of green objects (nucleus). Marked comparison indicates significantly different comparison (n = 3; $P$-value < 0.05, $t$ test assuming unequal variance).

2018). Last, Merrick et al (2019) identified distinct adipocyte progenitor populations from mouse adipose tissues, including interstitial progenitors and committed preadipocytes (Merrick et al, 2019). Comparative analysis of these subpopulations with our data showed that ASCs (our study) are similar to ASC 2 (Burl et al, 2018), FIP (Hepler et al, 2018), interstitial progenitor (Merrick et al, 2019), and G1 and G4 (Schwalie et al, 2018), whereas PAs in our study are similar to ASC 1 (Burl et al, 2018), APC and PA (Hepler et al, 2018), Icam1+ PA (Merrick et al, 2019), and G2 (Schwalie et al, 2018) (Fig 2E). Functional studies (differentiation and proliferation) reported by (Hepler et al, 2018) and Merrick et al (2019) further corroborated the comparative bioinformatics analysis: (1) like ASCs identified in the present study, FIPs had significantly less adipogenic differentiation capacity than APCs (Hepler et al, 2018); (2) interstitial progenitors had greater proliferation capacity but less adipogenic differentiation capacity than Icam1+ PAs (Merrick et al, 2019). Thus, the APC subpopulations described in the present study may represent similar or highly overlapping subpopulations to those identified in previously published work, although there is a limitation in direct comparison between these cells because of differences in sample preparation and/or functional characterization protocols. Of note, although the aforementioned scRNA-seq studies have shown that transcriptomically distinct APC subpopulations exist, transcriptomic profiles of these subpopulations across these studies had not been performed. Our study revealed that transcriptomically distinct APC subpopulations can be identified and prospectively isolated for downstream analyses. We further demonstrated that subpopulation proliferation/differentiation capacities are well correlated with multiple, independently curated transcriptomic profiles.

It is also noteworthy that the isolated APC subpopulations described in the present study have distinct differentiation capacities. In a recent scRNA-seq analysis of human APCs, data suggest that human APCs are a homogeneous population (Acosta et al, 2017). This is in contrast to the aforementioned murine APC scRNA-seq studies (Burl et al, 2018; Hepler et al, 2018; Schwalie et al, 2018; Merrick et al, 2019) as well as the present study, although it is important to note that the Acosta et al (2017) dataset profiled a limited number of progenitor cells (n = 381), thus making population heterogeneity difficult to assess. Of note, in support of human APC functional heterogeneity, a recent study by Raajendiran et al (2019) showed that APC subpopulations (stratified based on CD34 expression) exhibited distinct metabolic properties albeit with negligible differences in adipogenic potential (Raajendiran et al, 2019). Taken together, these studies highlight the complexity of APC heterogeneity, but collectively support the hypothesis that APCs are a functionally heterogeneous cell population. How each of these subpopulations contribute to adipose tissue homeostasis (or pathology) will be important to assess in future work.

### DIO results in APC subpopulation shifts and marked changes in subcluster gene expression

Previous studies investigating APC heterogeneity primarily focused on identifying and characterizing distinct subpopulations in healthy tissue (Burl et al, 2018; Hepler et al, 2018; Schwalie et al, 2018; Merrick et al, 2019). How DIO affects these APC subpopulations, however, is not known. As shown in Fig 1F, DIO rearranged the distribution of APC subpopulations. We next asked if cluster-specific gene expression was altered in obese APCs. Control and DIO subpopulations for each individual cluster were combined and NMF analyses performed. All five clusters were optimally separated by k = 2 (Figs 6A–F and S7), indicating that two subgroups (please note dark and light colors within each subcluster in Fig 6A–F) exist within each of the five APC clusters. With the exception of cluster 5, DIO resulted in a shift in the dominant subgroup within each cluster. We then compared cluster-specific subgroup pairs to find DE genes within each cluster in response to DIO (Tables S6–S10). We found widespread DIO-associated up-regulation of several ECM components, such as Col3a1, Postn, Thbs1, Col6a1, Col6a3, Col15a1, Col1a1, Bgn, Fbln1, Eln, Col5a3, Col4a1, Col4a2, and Col6a2. Among these factors, Col3a1, Col6a1, Col6a2, Col6a3, Bgn, and Thbs1 were commonly identified in multiple clusters, whereas other ECM components were identified in a cluster-specific manner. Many of these ECM components, including Col3a1, Postn, Thbs1, Col6a1, Col6a3, Col1a1, Bgn, Eln, Col4a1, and Col4a2, are reportedly up-regulated in obese adipose tissue (Table S11), whereas other ECM factors (Col15a1, Fbln1, Col5a3, and Col6a2) do not have a prior link to obesity.

In addition to ECM remodeling, DIO expanded APC subgroups expressing higher levels of several proinflammatory factors, such as Ccl2, Ccl7, and Cxcl1 in cluster 5 ASCs, and Ccl11 in cluster 6 PAs. Although these proinflammatory factors are not directly linked to obesity, their up-regulation is consistent with immunomodulatory abilities often attributed to stressed adipose-derived stem/stromal cells. Moreover, transcripts associated with obesity-induced adipose tissue inflammation, such as Col6a3, Thbs1, Bgn, Postn, and Cd44 (Table S11), were highly expressed in cluster subgroups that were more abundant in DIO conditions. Pathway analyses performed on cluster-specific subgroup DE genes confirmed these individual observations and identified numerous pathways involved in inflammation and immune response, such as PI3K signaling in B lymphocytes, leukocyte extravasation signaling, and dendritic cell maturation (Fig 6G).

Finally, transcripts linked to adipocyte function and adipogenic differentiation were differentially expressed between cluster-specific subgroups, including Meg3 (cluster 1), Klf2, Klf4, Jun, Fos, and Mgp (cluster 4), Lgals1 and Anxa2 (cluster 5), and S100a16 (cluster 6). Meg3, Klf2, Klf4, Lgals1, and S100a16 regulate adipocyte differentiation and increased expression of Mgp and S100a16 are associated with adipocyte differentiation (Table S11). Pathway analyses corroborated these observations and identified several signaling cascades, including Rho signaling and ILK signaling (Fig 6G), that negatively regulate adipocyte differentiation. Together, these data show that DIO leads to significant changes in APC subpopulation distribution, potentially enriching for APC subpopulations exhibiting enhanced ECM and immunomodulatory capabilities and altered differentiation capacities. Further studies are needed to reveal underlying mechanisms as to how these DIO-induced alterations in gene expression and APC subpopulation composition contribute to obesity-associated APC dysfunction.

### Conclusions

APCs are major contributors to obesity-associated increases in adipose tissue mass. In this study, we used single-cell sequencing

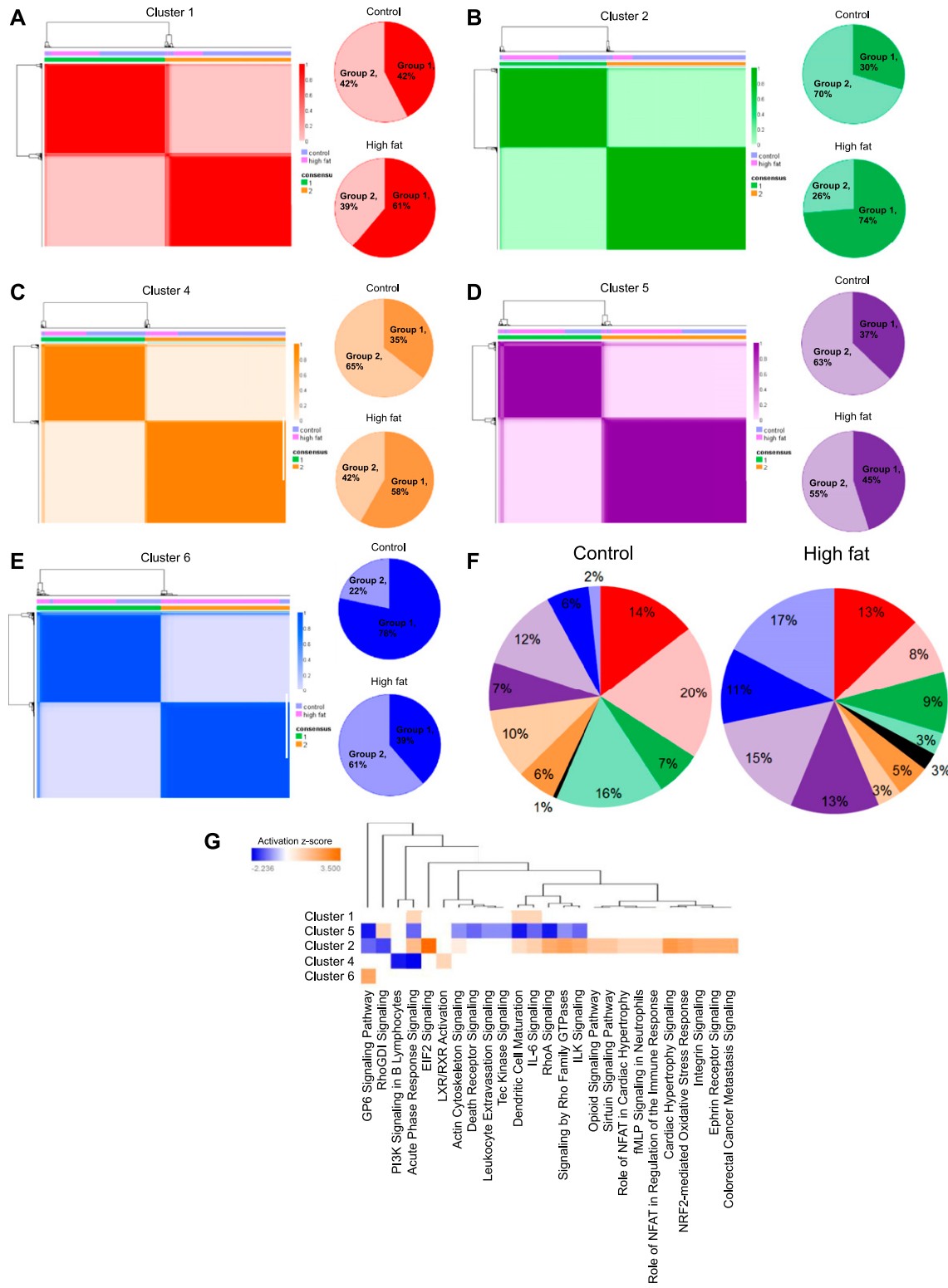

**Figure 6. Comparative analyses of control and high-fat/DIO APC subpopulations.**
**(A, B, C, D, E)** NMF clustering on cells in clusters 1, 2, and 4–6 from control and high-fat/DIO conditions. Subclusters are shown with color gradients in consensus matrices generated by NMF clustering (left), and their relative proportions in control and high-fat/DIO conditions are shown in pie charts (right). **(F)** A summary of the distribution of cells in each subcluster with respect to the overall APC pool. **(G)** Pathways associated with differentially expressed genes in group 2 versus group 1 of each cluster.

to identify and characterize APC subpopulations in healthy and obese visceral adipose tissue. We found that although APCs can be generally subdivided into adipose stem cells and preadipocytes, significant differences exist within each of these cell populations. APCs isolated from obese tissue exhibit marked differences in subpopulation distribution, pathway activation, and differentiation status. Isolation and comparison of these APC subpopulations between control and obese conditions will be necessary to further validate these transcriptomic observations. Moving forward, a more sophisticated understanding of how each of these subpopulations influence adipose tissue homeostasis may provide insights into the etiology of obesity, thus revealing novel strategies to limit obesity-associated APC dysfunction.

# Materials and Methods

### Animals

All procedures using animals in this study were approved by the Mayo Clinic Institutional Animal Care and Use Committee. 3–4-mo-old male Friend leukemia virus B (FVB) mice were used for all experiments in this study. For DIO and control diet feeding, 45% kcal fat diet (TD.08811; Envigo) and low glycemic control diet (TD.120455; Envigo) were used after feeding with control diet for 1 wk for acclimation.

### APC isolation and flow cytometry

APCs were isolated from epididymal fat pads as previously described (Joseph et al, 2018) with slight modifications as follows. After tissue dissociation with collagenase II (Thermo Fisher Scientific), the cells were filtered through a 70-$\mu$m cell strainer. The cells were washed once and resuspended in 45 $\mu$l flow cytometry buffer (PBS supplemented with 2-mM EDTA and 0.5% [wt/vol] bovine serum albumin [Gold Biotechnology]). 5 $\mu$l FcR blocking reagent (Miltenyi Biotec) was then added and incubated for 10 min at 4°C, followed by mixing with anti-Sca1-APC (1:10) (130-102-833; Miltenyi Biotec), anti-CD31-FITC (1:10) (130-102-519; Miltenyi Biotec), anti-CD45-FITC (1:10) (130-102-491; Miltenyi Biotec), and anti-Ter119-FITC antibodies (1:10) (130-112-908; Miltenyi Biotec). The cells were incubated for 10 min at 4°C. Then, the cells were diluted with 500 $\mu$l flow cytometry buffer supplemented with 1 $\mu$g/ml DAPI. DAPI[neg]/Sca1[pos]/CD31[neg]/CD45[neg]/Ter119[neg] cells were sorted by FACSAria instrument (BD Biosciences) with indicated gating strategies at Microscopy Cell Analysis Core Flow Cytometry Facility in Mayo Clinic.

Flow cytometry analyses were performed using a MACSQuant instrument (Miltenyi Biotec), and data analyzed using FlowJo software (BD Biosciences). To isolate subpopulations of APCs, flow-sorted APCs (DAPI[neg]/Sca1[pos]/CD31[neg]/CD45[neg]/Ter119[neg]) were co-stained with anti-CD81-VioGreen (1:10) (130-108-390; Miltenyi Biotec), anti-CD55-PE-Vio770 (1:10) (130-104-054; Miltenyi Biotec), and anti-CD9-VioBlue antibodies (130-103-384; Miltenyi Biotec) for 10 min at 4°C. The cells were then diluted with flow cytometry buffer supplemented with propidium iodide solution (1:100) (130-093-233; Miltenyi Biotec). APC subpopulations were sorted as described above.

### scRNA-seq and data analysis (NMF, IPA)

APCs were isolated as described above from a pool of four mice after 2 wk of special diet feeding. Single-cell capture and cDNA library preparation was performed on isolated APCs using Chromium Single Cell 3′ Reagent Kit v2 (10× Genomics) according to the manufacturer's protocol. cDNA libraries were sequenced on an Illumina HiSeq 2500 platform (Illumina). To minimize batch effects between samples, cell preparation and scRNA-seq procedures were performed simultaneously for all samples. All of these procedures were performed in collaboration with the Mayo Clinic Medical Genome Facility.

Sequencing outputs were aligned to mm10 and processed by Cell Ranger 2.2.0 pipeline (10× Genomics) in collaboration with Mayo Clinic Bioinformatics Core. The data output with expression levels of 27,998 genes was normalized and further analyzed in R software for PCA, tSNE (van der Maaten & Hinton, 2008), and k-means clustering using run_pca, run_tsne, and run_keans_clustering commands in "cellrangerRkit" package (10× Genomics), respectively. For further downstream analysis, genes with expression levels of zero in all cells were removed to retain 17,184 genes. Hierarchical clustering was performed with complete linkage method, and distances were measured by correlation, using TIBCO Spotfire software. Differentially expressed genes were identified by SAM (Tusher et al, 2001) using "samr" package in R software. NMF (Brunet et al, 2004) was performed using "NMF" package in R. Pathway analysis was performed on differentially expressed genes (fold change >1.5 and q-value = 0) using Ingenuity Pathway Analysis (QIAGEN).

### Comparative single-cell expression profiling meta-analysis

The following published scRNA-seq datasets from mouse adipose tissues were downloaded with these accession IDs: SRX4074084, SRX4074085, SRX4074088, and SRX4074089 from Sequence Read Archive (Burl et al, 2018); GSE111588 from Gene Expression Omnibus (Hepler et al, 2018); E-MTAB-6677 from ArrayExpress (Schwalie et al, 2018); and GSE128889 from Gene Expression Omnibus (Merrick et al, 2019). Each of these datasets was processed and analyzed to reproduce the published results using reported methods in each article using Cell Ranger pipeline (10× Genomics) and "Seurat" package in R software. After cell populations were determined, adipogenic progenitor populations in the datasets were retained for comparison. The datasets were normalized using NormalizeData command. Then, the datasets were integrated with batch-correction using FindIntegrationAnchors and IntegrateData commands in "Seurat" package. Combined data were visualized on tSNE plots using ScaleData, RunPCA, and RunTSNE commands.

### Cell culture and ex vivo expansion and differentiation assays

Ex vivo proliferation assays were performed as previously described (Joseph et al, 2018). Briefly, each APC subpopulation was plated into multiple wells of a 96-well tissue culture plate at 1,000 cells/well in DMEM supplemented with 10% FBS, 100 U/ml penicillin–streptomycin, and 0.01 $\mu$g/ml human FGF2 (Gold Biotechnology). Differentiation was induced immediately after each APC subpopulation was cultured to full confluence. Because each APC subpopulation reached to full

confluence at different time points because of distinct proliferation rates, cell confluence in each subpopulation was monitored every 4 h during cell culture by an IncuCyte ZOOM (Essen BioSciences). Differentiation was induced by replacing media with differentiation media consisting of DMEM, 4 nM insulin (Sigma-Aldrich), 1 $\mu$M rosiglitazone (Sigma-Aldrich), 10% FBS, and 100 U/ml penicillin–streptomycin. To monitor and assess differentiation, 7.5 $\mu$M BODIPY 558/568 C12 (Thermo Fisher Scientific) was added during differentiation. All cell culture was performed with an IncuCyte ZOOM (Essen BioSciences) to monitor and quantify cell confluence and red fluorescence (BODIPY 558/568 C12) to assess cell proliferation and differentiation. After 10 d of induction of differentiation in each APC subpopulation, the cells were fixed in 4% paraformaldehyde, and differentiation was assessed by staining with 7.5 $\mu$M BODIPY 558/568 C12 (Thermo Fisher Scientific) and 3.3 $\mu$M Nuclear Green DCS1 (Abcam). Cell confluence, lipid area (red fluorescence area), and the number of nuclei (the number of green fluorescence objects) of at least three areas in each well were quantified using IncuCyte ZOOM 2016B software (Essen BioSciences).

### Quantitative RT-PCR

Total RNA was isolated from samples using RNeasy Plus Mini Kit (QIAGEN), followed by cDNA synthesis using High-Capacity cDNA Reverse Transcription Kit (Thermo Fisher Scientific). Transcript levels were then quantified with SsoAdvanced Universal SYBR Green Supermix (Bio-Rad) by CFX384 Touch thermal cycler (Bio-Rad). All of these procedures were performed according to the manufacturers' protocols. Primer sequences are shown in Table S12.

## Data Availability

Datasets generated during the current study are available at Sequence Read Archive under accession number SRP226152.

## Supplementary Information

## Acknowledgements

The authors wish to thank members of the Doles lab for helpful discussions and manuscript suggestions. JD Doles was supported by National Institute of Arthritis and Muscoloskeletal and Skin Diseases AR66696, National Institute of General Medical Sciences GM128594, Mayo Clinic start-up funds, and the Glenn Foundation for Medical Research. Single cell studies were made possible through collaboration with the Mayo Clinic Medical Genome Facility.

### Author Contributions

DS Cho: conceptualization, data curation, formal analysis, investigation, methodology, and writing—original draft, review, and editing.
B Lee: data curation, formal analysis, and investigation.

JD Doles: conceptualization, formal analysis, supervision, funding acquisition, project administration, and writing—original draft, review, and editing.

### Conflict of Interest Statement

The authors declare that they have no conflict of interest.

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
