## [Reviewer comments · Life Science Alliance]

Life Science Alliance

Refining the adipose progenitor cell landscape in healthy and obese visceral adipose tissue

Dong Cho, Bolim Lee, and Jason Doles
DOI: <https://doi.org/10.26508/lsa.201900561>

Corresponding author(s): Jason Doles, Mayo Clinic

Review Timeline:	Submission Date:	2019-09-20
	Editorial Decision:	2019-09-23
	Revision Received:	2019-10-20
	Editorial Decision:	2019-11-06
	Revision Received:	2019-11-13
	Accepted:	2019-11-14

Scientific Editor: Andrea Leibfried

Transaction Report:

Please note that the manuscript was previously reviewed at another journal and the reports were taken into account in the decision-making process at Life Science Alliance.

Referee #1 Review

Report for Author:

The study by Cho et al. uses single cell transcriptomics to monitor adipose precursor cell (APC) heterogeneity (based on an analysis of mouse epididymal fat) and its putative compositional change in an obesogenic context. Several APC subpopulations were identified, which, upon isolation, exhibited distinct differentiation properties. Furthermore, the authors found that the proportion of several of these subpopulations changes upon excessive weight gain (using a high fat diet), with expanding subpopulations showing increased expression of genes linked to ECM remodeling and immune-modulation.

In sum, while the biological conclusions of this study, including the fact that adipose tissue is plastic, are not surprising, the presented data seems of high quality and has thus value as a resource for the field.

Nevertheless, the study could be improved in many ways, as detailed below:

- 1) The authors make adipose tissue-wide conclusions, yet their analysis is only based on analysis of one particular fat depot: epididymal (visceral) fat. While an analysis of other fat depots may be outside the scope of this study, they should at least make this very clear throughout the manuscript, including the title and abstract.
- 2) The single cell data are presented in a convoluted way and it is often difficult to follow the authors' discussion on the different clusters:
 - a. First, it is rather hard to judge the final quality of the data as the methodological description of the analyses is minimal. For example, how has the quality check been performed? Except for cluster 3, have other cells been filtered? Which genes have been filtered beyond those that are not expressed in any cell? On which set of genes has the tSNE been generated or has the clustering been performed (Fig. 2A)? Even if it appears as if there is no real batch effect, it is unclear if the authors designed the experiment to limit batch effects and if they used any batch correcting tool? Has any gene filtering been applied to perform the differential gene expression analyses? Etc etc. In short, the authors need to substantially expand the scRNA-seq methods section of the paper such that independent readers can reproduce their analyses.
 - b. The usual approach to select the number of clusters with kmeans clustering is to calculate the clusters within the sum of square and to generate an elbow plot. However, the fashion by which the authors of this study selected 6 clusters is rather unusual and should therefore be revisited to evaluate how robust the existence of the reported six clusters truly is. It would thereby also be helpful if the clusters were renamed with intuitive names and with a color code that would be maintained all along the manuscript.

c. It would be most helpful if the authors would present their single cell data in the context of established literature on mouse APC heterogeneity. In particular, both Schwalie et al., Nature, 2018 and Merrick et al., Science, 2019 clearly demonstrated the presence of three adipose APC subpopulations (from subcutaneous inguinal WAT). It would in this regard be highly valuable if the authors could project their data on these established clusters (e.g. using scmap or other tools) to evaluate whether the detected clusters for the most part overlap these previously reported subpopulations or whether new heterogeneity has been uncovered. A cursory view on differentiating genes suggest the former, but a more in-depth analysis would be required to address this important question.

3) The authors suggest that the proportion of subpopulations is altered in obese mice, yet base this conclusion solely on their scRNA-seq data. Given the technical issues that could occur when isolating and processing individual cells for transcriptomics, the authors should at least validate their findings using an orthogonal method, for example using flow cytometry based on the markers that they identified (CD55, CD81, and CD9). Note also the error (lines 213-214) describing cluster 1 as CD55^{high} / CD81^{high} and cluster 5 as CD55^{high} / CD81^{low}, whereas, based on Fig.3 it is the opposite.

4) The methodological description of how adipocyte differentiation was quantified is also minimal and should be expanded for independent readers to be able to reproduce these results. Representative images for each cluster should also be shown. Furthermore, the use of rosiglitazone is not common practice in the field to assess differentiation potential. More standard is the use of a differentiation cocktail consisting of IBMX, dexamethasone and insulin, which should be used here as well to enable a more objective comparison of the behavior of the tested subpopulations to already published observations in the field.

5) Making the datasets of this study only available "upon request" is simply not acceptable and this paper should not be accepted for publication prior to proper depositing of the data in public databases such as SRA.

6) The authors state that previous scRNA-seq studies of APC heterogeneity have not rigorously assessed the functional properties of the detected subpopulations. This is false as for example both the Merrick et al. and Schwalie et al. papers have used an arsenal of functional tests (differentiation assays, transplants, lineage tracing, transwells etc.) to phenotype the detected subpopulations. In light of the limited downstream experiments presented in this study, this statement should be removed.

Referee #2 Review

Report for Author:

In their manuscript "Obesity alters the adipose progenitor cell landscape" Cho, Lee and Doles describe the perigonadal adipose tissue precursor composition from lean and diet induced obese mice. The authors perform single cell RNA sequencing of more than 2000 FACS sorted precursor cells and mainly focus their analysis on gene expression differences between the identified clusters and how cellular distribution and gene expression within the clusters changes upon high fat diet induced obesity. Moreover, the authors describe a strategy to sort five different precursor populations. Applying this strategy the authors further demonstrate differences in proliferation and differentiation capacity of these different cell populations.

Overall the manuscript is very well written and provides an additional view on perigonadal adipose tissue precursor heterogeneity, on top of what has been previously published by others. However, it is not clear to me what the advancement of this study over the, from the authors cited, previous studies is. It would be very interesting to perform this direct comparison between the authors' data and previously published data sets to take full advantage of the increased sequencing depth by the authors. Moreover, the description of the methods for determining cellular proliferation and differentiation are not detailed enough to fully judge these results. I assume that for the differentiation the authors waited until all cells in the plate reached confluency. However, in that case, cells with a higher proliferation might have been too crowded, which could result in impaired adipogenesis. Lastly, I would recommend to provide some kind of functional assays describing differences in the adipocytes derived from these distinct precursor pools, or at the very least demonstrate some biological meaning for the existence of these subpopulations.

Additional comments:

It would be good to provide references for the genes used to show certain features, such as differentiation through Rho and ILK e.g..

Fig. 3B: Cluster 1 does not seem to show a discrete population

Fig. 3c-f: what is the n?

Fig 3&4: T-tests seem inappropriate for comparison. Please use either one- or two-way ANOVA with the appropriate post-hoc tests for these multiple comparisons.

Referee #3 Review

Report for Author:

Cho et al presented a study where they analysed changes in gene expression profiles of pan 'adipose progenitors' from mouse epididymal fat pads from normal and obese mice at single cell level. They identified 6 clusters, where cluster 1 and 5 are the more primitive ASCs, clusters 2, 4, 6 are 'preadipocytes', while cluster 3 is a contamination/artefact of sample preparation. They identified FACS strategies to isolate these populations and demonstrated that the FACS sorted populations differed in the ability to proliferate and differentiate in culture. They also described the alteration of clusters distribution by diet-induced obesity.

In general, the study is touching an important and interesting topic in the field of adipose tissue biology. The manuscript is fairly well-written, although more explanations behind the bioinformatics analysis will help the understanding of the manuscript. However, there are several major issues that needed to be addressed, and they are as follows:

Major points

- 1) Novelty. There are several comprehensive single cell RNA-seq papers in adipose tissue published, where clusters/novel populations and markers are identified, and the effect of DIO on gene expression/ population is also included in these studies. The novelty aspect of this manuscript needs to be addressed.
- 2) Consolidate current data with published/existing data. The authors sequenced cells from adult epididymal fat pad but did not seem to obtain identical result compared with published studies. There is a brief paragraph in the manuscript discussing previous published studies, but this is not enough. It is necessary to compare current data with published single cell RNA-seq data.
- 3) The evidence suggests that there are distinctive clusters needs strengthening. Could the authors present marker gene expression patterns across the clusters, highlighting the expression patterns of selected genes (e.g. tSNE plots of known ASC markers, preadipocyte markers, and cluster-specific marker genes)
- 4) It is very nice to see authors have FACS sorted the populations of cells and studied their behaviour in culture as well as validated gene expression of the sorted cells by QPCR analysis. However, the rationale of FACS markers chosen and data presented in the FACS gating are not convincing, and there are not enough QPCR data to fully support that the FACS strategy employed has indeed separated these clusters. The concerns are as follows:
 - CD81: Fig3A, the levels of CD81 of clusters 1/5 and 2/4/6 need to be presented in the same plot (unless the Y-axis of the two separate CD81 plots in Fig 3A are directly comparable).
 - Cluster 2, 4, 6 are sorted first based on CD9 v.s. CD55, then by Cd81. Would the same data be obtained by sorting via CD81 v.s. CD55, then separated by CD9?
 - QPCRs results are required to demonstrate CD55, CD9, and CD81 levels in the sorted populations. The authors have measured CD55 (Fig 3C); however the result

indicated that the level of Cd55 in cluster 1 does not differ from cluster 2, and cluster 5 does not differ from cluster 2.

- There are typos describing cluster 1 and 5 (in the text page 9).
- What are the levels of CD9 in cluster 1 and 5?
- How are these markers chosen and how do they relate to published isolation strategy published by other single cell studies?

5) Assaying of differentiation potential between sorted cells. The authors have demonstrated nicely that cluster 1, 5, and 6 proliferate at a faster rate than cluster 2 and 4. The differentiation ability was performed by averaging lipid area over 'green objects' which indicate cell number. Wouldn't this approach make clusters 1, 5, 6 have smaller differentiation potential simply because there are more cells in the well? This makes up a major claim of the current study and more evidence demonstrating clusters 2 and 4 indeed differentiate better is required (e.g. QPCRs analysis of differentiation markers between clusters, and images demonstrating the different differentiation potential between clusters. Would the image in Fig 4B suggest that majority of the sorted APCs did not differentiate?). In addition, if the data were acquired using IncuCyte, were there images taken at multiple time points? It will make a stronger argument to present data from different time points (if there are data acquired already).

Minor points

- 1) The title emphasises on obesity induced alteration to adipose progenitor cell distribution, but most the study focuses on the populations/clusters obtained from normal diet. The last section about DIO results in APC subpopulation shifts etc needs clarification.
- 2) Fig 5 and the relating text are really difficult to follow. In addition, obesity induced changes in adipose tissue are well known, but there is no discussion of the data obtained in the current study with published work (i.e. what is new and for broad biological significance?)
- 3) More explanations need to be given behind the bioinformatics analysis.
- 4) There is no indication of the number of biological replicates performed in some experiments (eg. QPCRs).
- 5) Fig 3C-F, do the results indicate that most of the comparisons of gene expression between clusters are not significant? (The error bars are large)
- 6) Fig 2G is missing
- 7) Explanation of the result in Supp Fig 5G is not clear.
- 8) Should give clear definitions of ASC, PA, and APCs that are used in this study.
- 9) This study suggested that clusters 1 and 5 are more 'primitive' than clusters 2, 4, and 6 and hence referred clusters 1 and 5 as ASCs. Do the chondrogenic and osteogenic differentiation abilities differ between clusters 1/5 and clusters 2/4/6? (It is not necessary to do these experiment, a discussion is sufficient)
- 10) Could the authors comment on the location of these clusters in adipose tissue? (especially clusters 1/5)
- 11) Do cluster 1/5 and clusters 2/4/6 possess hierarchical relationships? (i.e. do

clusters 1/5 give rise to clusters 2/4/6?) It is not essential to demonstrate this by experiments as this is not trivial, discussions would be sufficient.

September 23, 2019

Re: Life Science Alliance manuscript #LSA-2019-00561-T

Dr. Jason Doles
Mayo Clinic
Guggenheim 16-11A1
200 First Street SW
Rochester, MN 55905

Dear Dr. Doles,

Thank you for transferring your manuscript entitled "Obesity alters the adipose progenitor cell landscape" to Life Science Alliance. Your manuscript was assessed by expert reviewers at another journal before, and the editors transferred those reports to us with your permission.

The reviewers appreciated your data but would have expected a further reaching advance. This concern does not preclude publication in Life Science Alliance and we would thus like to invite you to submit a revised version of your manuscript to us, based on the reviewer reports obtained at the other journal. We would expect a point-by-point response to all concerns raised and accordingly changes to manuscript text & data representation as well as a re-analysis of the data already at hand and the requested extension to include a comparative analysis. More specifically, please:

Rev#1, points:

- 1 - address in text
- 2a - add information and clarify
- 2b - revisit with data already at hand
- 2c - perform comparative analysis
- 3 - acknowledge in text
- 4 - discuss in text
- 5 - please address as requested by reviewer
- 6 - address in text

Rev#2:

general statement: respond in point-by-point response
additional comments: address

Rev#3, points:

- 1 - address in text
 - 2 - perform comparative analysis
 - 3 - address
 - 4 - clarify / respond in point-by-point response
 - 5 - address in text and rebuttal
- address minor points

You will be guided to complete the submission of your revised manuscript and to fill in all necessary information. Your login name is jdoles2129

Thank you for this interesting contribution to Life Science Alliance. We are looking forward to receiving your revised manuscript.

Sincerely,

B. MANUSCRIPT ORGANIZATION AND FORMATTING:

Dear Dr. Leibfried and *Life Science Alliance* editorial staff:

Thank you for considering our manuscript and assisting us in prioritizing key experiments for revision. We are pleased to submit a revised manuscript that addresses all of your questions/concerns. We are excited to see that the manuscript was well received overall and agree with reviewers that it has the potential to significantly impact/add to the field of adipose progenitor cell biology. Below, please find our point-by-point response (our answers in **bold red** below) to reviewer concerns:

Referee #1:

The study by Cho et al. uses single cell transcriptomics to monitor adipose precursor cell (APC) heterogeneity (based on an analysis of mouse epididymal fat) and its putative compositional change in an obesogenic context. Several APC subpopulations were identified, which, upon isolation, exhibited distinct differentiation properties. Furthermore, the authors found that the proportion of several of these subpopulations changes upon excessive weight gain (using a high fat diet), with expanding subpopulations showing increased expression of genes linked to ECM remodeling and immune-modulation.

In sum, while the biological conclusions of this study, including the fact that adipose tissue is plastic, are not surprising, the presented data seems of high quality and has thus value as a resource for the field.

We thank the reviewer for the positive assessment of our data quality and potential value to the field.

Nevertheless, the study could be improved in many ways, as detailed below:

1) The authors make adipose tissue-wide conclusions, yet their analysis is only based on analysis of one particular fat depot: epididymal (visceral) fat. While an analysis of other fat depots may be outside the scope of this study, they should at least make this very clear throughout the manuscript, including the title and abstract.

We apologize for the overstatements and agree that our conclusions should be limited to visceral fat APC dynamics. Text (and title) changes were made in accordance with this recommendation.

2) The single cell data are presented in a convoluted way and it is often difficult to follow the authors' discussion on the different clusters:
a. First, it is rather hard to judge the final quality of the data as the methodological description of the analyses is minimal. For example, how has the quality check been performed? Except for cluster 3, have other cells been filtered? Which genes have been filtered beyond those that are not expressed in any cell? On which set of genes has the tSNE been generated or has the clustering been performed (Fig. 2A)? Even if it appears as if there is no real batch effect, it is unclear if the authors designed the experiment to limit batch effects and if they used any batch correcting tool? Has any gene filtering been applied to perform the differential gene expression analyses? Etc etc. In short, the authors need to substantially expand the scRNA-seq methods section of the paper such that independent readers can reproduce their analyses.

Thank you for pointing out these method description deficiencies. We have revised the methods section accordingly. In brief and with respect to batch effect, all of our

procedures for single-cell RNA-sequencing (scRNA-seq) were performed simultaneously for all samples to minimize this issue. In addition, batch effects during the comparison with ours and other published scRNA-seq datasets were corrected using 'FindIntegrationAnchors' and 'IntegrateData' commands in 'Seurat' package which corrects batch effects between different datasets.

b. The usual approach to select the number of clusters with kmeans clustering is to calculate the clusters within the sum of square and to generate an elbow plot. However, the fashion by which the authors of this study selected 6 clusters is rather unusual and should therefore be revisited to evaluate how robust the existence of the reported six clusters truly is. It would thereby also be helpful if the clusters were renamed with intuitive names and with a color code that would be maintained all along the manuscript.

We agree with the reviewer that, as presented, our assignment of six clusters seemed arbitrary. In addition to revising the text to better articulate our methodology, we have reanalyzed the data and added an elbow plot plotting total within-clusters sum of squares against number of clusters (k) (new data added to Figure S2). The elbow plot indicates that 7 clusters would be the most appropriate to cluster our data. However, for k=7 (and as supported by our prior data in Fig S2), the number of cells in one cluster (see blue dots in tSNE plot for k=7 in Figure S2) was too low to define and compare them between control and high-fat feeding conditions. Furthermore, aside from this minimal cluster, the remaining 6 clusters clustered nearly identically to those for k=6 (Figure S2C). Hence, we chose k=6 as the optimal cluster number to define each of these subpopulations in the downstream analysis. These points were addressed in the text as well. We also clarified the text/figures to emphasize that clusters 1/5 best resemble ASCs (our own analysis as well as the comparative analyses) and that clusters 2/4/6 best resemble pre-adipocytes/more mature adipose progenitors. The naming and coloring scheme has also been updated to maintain consistency throughout the manuscript.

c. It would be most helpful if the authors would present their single cell data in the context of established literature on mouse APC heterogeneity. In particular, both Schwalie et al., Nature, 2018 and Merrick et al., Science, 2019 clearly demonstrated the presence of three adipose APC subpopulations (from subcutaneous inguinal WAT). It would in this regard be highly valuable if the authors could project their data on these established clusters (e.g. using scmap or other tools) to evaluate whether the detected clusters for the most part overlap these previously reported subpopulations or whether new heterogeneity has been uncovered. A cursory view on differentiating genes suggest the former, but a more in-depth analysis would be required to address this important question.

We agree that a comparative analysis is warranted. To address this point, we compared our data to four other scRNA-seq datasets for adipose tissues/adipose tissue progenitor cells (Burl et al., 2018; Hepler et al., 2018; Merrick et al., 2019; Schwalie et al., 2018). Data are presented in Figure 2E. Comparative analyses suggest a high degree of concurrence with prior published data, but also highlight additional heterogeneity within these previously identified APC subpopulations, a point that we have added to/clarified in the text.

3) The authors suggest that the proportion of subpopulations is altered in obese mice, yet base this conclusion solely on their scRNA-seq data. Given the technical issues that could occur when isolating and processing individual cells for transcriptomics, the authors should at least validate their findings using an orthogonal method, for example using flow cytometry based on

the markers that they identified (CD55, CD81, and CD9). Note also the error (lines 213-214) describing cluster 1 as CD55high / CD81high and cluster 5 as CD55high / CD81low, whereas, based on Fig.3 it is the opposite.

We acknowledge the point that validation of our findings from transcriptomic data will highly support our conclusions on the relevance of the identified cell populations in an obesity setting. We are definitely planning to continue our work on the functional relevance of these subpopulations in various pathologies (obesity, aging, wasting, etc.) in the future, but we believe that these studies are out of the scope of this manuscript where our aim was to identify and provide proof-of-principle evidence that these sub-clusters are functionally distinct.

With respect to the CD55/CD81 issue, this was a clerical error and appropriate changes were made to the text/figures as appropriate.

4) The methodological description of how adipocyte differentiation was quantified is also minimal and should be expanded for independent readers to be able to reproduce these results. Representative images for each cluster should also be shown. Furthermore, the use of rosiglitazone is not common practice in the field to assess differentiation potential. More standard is the use of a differentiation cocktail consisting of IBMX, dexamethasone and insulin, which should be used here as well to enable a more objective comparison of the behavior of the tested subpopulations to already published observations in the field.

We appreciate the point regarding the differentiation protocol. The methods section was clarified appropriately, and we have acknowledged in the text that our observations have limitations. Further studies are ongoing to more rigorously compare the functional differences between these subpopulations, but these are outside of the scope of the present study. Lastly, representative images for each subpopulation for the differentiation assays are now shown in the main figure.

5) Making the datasets of this study only available "upon request" is simply not acceptable and this paper should not be accepted for publication prior to proper depositing of the data in public databases such as SRA.

We have deposited our data in SRA and text was changed accordingly.

6) The authors state that previous scRNA-seq studies of APC heterogeneity have not rigorously assessed the functional properties of the detected subpopulations. This is false as for example both the Merrick et al. and Schwalie et al. papers have used an arsenal of functional tests (differentiation assays, transplants, lineage tracing, transwells etc.) to phenotype the detected subpopulations. In light of the limited downstream experiments presented in this study, this statement should be removed.

We appreciate the point, and the sentence was removed and text was changed accordingly.

Referee #2:

In their manuscript "Obesity alters the adipose progenitor cell landscape" Cho, Lee and Doles describe the perigonadal adipose tissue precursor composition from lean and diet induced obese mice. The authors perform single cell RNA sequencing of more than 2000 FACS sorted

precursor cells and mainly focus their analysis on gene expression differences between the identified clusters and how cellular distribution and gene expression within the clusters changes upon high fat diet induced obesity. Moreover, the authors describe a strategy to sort five different precursor populations. Applying this strategy the authors further demonstrate differences in proliferation and differentiation capacity of these different cell populations. Overall the manuscript is very well written and provides an additional view on perigonadal adipose tissue precursor heterogeneity, on top of what has been previously published by others. However, it is not clear to me what the advancement of this study over the, from the authors cited, previous studies is. It would be very interesting to perform this direct comparison between the authors' data and previously published data sets to take full advantage of the increased sequencing depth by the authors.

We thank the reviewer for the overall positive assessment of our manuscript. As noted in the response to reviewer 1, we performed the requested comparative analysis to better support our conclusions. In brief, we combined our data with other scRNA-seq datasets from Burl et al., 2018, Hepler et al., 2018, Schwalie et al., 2018, and Merrick et al., 2019, in order to more directly compare our subpopulations with published work (new data shown in Figure 2). The comparative analysis shows that our ASC populations are similar to less adipogenic or stem cell specific populations in others' studies, while our PA populations are similar to more committed populations in others' studies. Importantly, our data further show additional heterogeneity within these APC subpopulations.

Moreover, the description of the methods for determining cellular proliferation and differentiation are not detailed enough to fully judge these results. I assume that for the differentiation the authors waited until all cells in the plate reached confluency. However, in that case, cells with a higher proliferation might have been too crowded, which could result in impaired adipogenesis. Lastly, I would recommend to provide some kind of functional assays describing differences in the adipocytes derived from these distinct precursor pools, or at the very least demonstrate some biological meaning for the existence of these subpopulations.

We have modified the text in our methods section to clarify and describe our functional studies in more detail. In brief, to maintain consistency and account for differences in proliferation rates between subpopulations, differentiation was induced right after each subpopulation reached full confluence. This, differentiation was induced at different timepoints for each subpopulation. We were able to precisely monitor subpopulation confluence (in real time) with an IncuCyte ZOOM, so the timing of differentiation could be determined and results subsequently compared.

Additional comments:

It would be good to provide references for the genes used to show certain features, such as differentiation through Rho and ILK e.g..

All references for the genes that we have described in this study are included in Table S12.

Fig. 3B: Cluster 1 does not seem to show a discrete population

We agree that clusters 1 and 5 (CD55 high) do not show discrete populations based on CD81 expression. This is likely because both cluster 1 and 5 are both expressing Cd81 (Figure 3A), although Cd81 is differentially expressed in the two clusters. However, CD81

expression was clearly separated into high and low in the CD55 low population. We therefore leveraged this difference to gate CD81 lower (cluster 1) and CD81 higher expressing cells (cluster 5) within the CD55 high population.

Fig. 3c-f: what is the n?

We apologize for the oversight. N=3, and it was added in the figure legend.

Fig 3&4: T-tests seem inappropriate for comparison. Please use either one- or two-way ANOVA with the appropriate post-hoc tests for these multiple comparisons.

We appreciate this comment and have amended the text to better justify our statistical analysis. In brief, ANOVA would be appropriate to find whether our measurements were differential overall, but the intent of the comparisons in Figures 3 and 4 was to test the [separate and] individual differences between each pair among our 5 clusters. In this way, pair-wise t-tests permit the detection of any potential differences in gene expression and proliferation/differentiation within a given pair of clusters.

Referee #3:

Cho et al presented a study where they analysed changes in gene expression profiles of pan 'adipose progenitors' from mouse epididymal fat pads from normal and obese mice at single cell level. They identified 6 clusters, where cluster 1 and 5 are the more primitive ASCs, clusters 2, 4, 6 are 'preadipocytes', while cluster 3 is a contamination/artefact of sample preparation. They identified FACS strategies to isolate these populations and demonstrated that the FACS sorted populations differed in the ability to proliferate and differentiate in culture. They also described the alteration of clusters distribution by diet-induced obesity.

In general, the study is touching an important and interesting topic in the field of adipose tissue biology. The manuscript is fairly well-written, although more explanations behind the bioinformatics analysis will help the understanding of the manuscript. However, there are several major issues that needed to be addressed, and they are as follows:

Major points

1) Novelty. There are several comprehensive single cell RNA-seq papers in adipose tissue published, where clusters/novel populations and markers are identified, and the effect of DIO on gene expression/ population is also included in these studies. The novelty aspect of this manuscript needs to be addressed.

We apologize for our insufficient discussion of study novelty in the previously submitted work. As explained above in our response to reviewers 1 and 2, we acknowledge that there are several scRNA-seq studies on adipose progenitor cells, and we have combined our dataset with these scRNA-seq datasets (Burl et al., 2018, Hepler et al., 2018, Schwalie et al., 2018, and Merrick et al., 2019) to better integrate/align our data with published work and to support our conclusions. We identified existing equivalents in adipose progenitor populations in accordance with other studies. With respect to novelty, we additionally identified the existence of further heterogeneity within these subpopulations by virtue of distinct transcriptomic profiles and proliferation/differentiation potential. This heterogeneity is also shown against the backdrop of obesity. The text was modified accordingly to highlight novel aspects of this work.

2) Consolidate current data with published/existing data. The authors sequenced cells from

adult epididymal fat pad but did not seem to obtain identical result compared with published studies. There is a brief paragraph in the manuscript discussing previous published studies, but this is not enough. It is necessary to compare current data with published single cell RNA-seq data.

As described above, the comparison with our data with published scRNA-seq data was performed, and results and text was modified accordingly. We thank the reviewers for this suggestion as the comparisons, overall, strengthen our manuscript and highlight how our data/analyses advance the previously published work.

3) The evidence suggests that there are distinctive clusters needs strengthening. Could the authors present marker gene expression patterns across the clusters, highlighting the expression patterns of selected genes (e.g. tSNE plots of known ASC markers, preadipocyte markers, and cluster-specific marker genes)

We show violin plots for differentially expressed canonical markers in Figure 2C (ASC markers), Figure 2D (PA markers). As requested, tSNE plots of these and other markers for ASC, PA, and each cluster are provided in a new Figure S5.

4) It is very nice to see authors have FACS sorted the populations of cells and studied their behaviour in culture as well as validated gene expression of the sorted cells by QPCR analysis. However, the rationale of FACS markers chosen and data presented in the FACS gating are not convincing, and there are not enough QPCR data to fully support that the FACS strategy employed has indeed separated these clusters. The concerns are as follows:

- CD81: Fig3A, the levels of CD81 of clusters 1/5 and 2/4/6 need to be presented in the same plot (unless the Y-axis of the two separate CD81 plots in Fig 3A are directly comparable).

The figure was corrected, plotting Cd81 expression in cluster 1/2/4/5/6 in the same plot.

- Cluster 2, 4, 6 are sorted first based on CD9 v.s. CD55, then by Cd81. Would the same data be obtained by sorting via CD81 v.s. CD55, then separated by CD9?

The two strategies yielded similar results. However, it was easier to gate first on CD9 vs. CD55 because CD9 (VioBlue) signal was higher overall than CD81 (VioGreen) and it was easier to draw gating for CD9 high population first in CD9 vs. CD55.

- QPCRs results are required to demonstrate CD55, CD9, and CD81 levels in the sorted populations. The authors have measured CD55 (Fig 3C); however the result indicated that the level of Cd55 in cluster 1 does not differ from cluster 2, and cluster 5 does not differ from cluster 2.

Cd9 and Cd81 were not among the top differentially expressed genes in clusters 1/5 vs. 2/4/6, although they are differentially expressed between clusters 1 and 5, and between clusters 2/4 and 6, respectively. Hence, CD9 and CD81 were used to isolate these clusters, and we did not confirm expression of these genes when comparing clusters 1/5 vs. 2/4/6. We also acknowledge that Cd55 expression did not pass the threshold for significantly differential expression in cluster 1 vs. 2 and in cluster 5 vs. 2, although its expression in cluster 2 is moderately lower than cluster 1 and 5. This is likely because we rely on transcriptomic data (RNA) to select surface marker proteins, and because the FACS-sorted subpopulations were based on expression of a limited number (3) of proteins, this may cause a slight discrepancy between transcriptomically distinct

populations and FACS-sorted populations. We strived, however, to confirm the differential expression of other, more robust transcripts to validating the surface marker-based strategy to separate these newly identified APC sub-populations. This validation is highlighted in Figure 4.

- There are typos describing cluster 1 and 5 (in the text page 9).

The typos were corrected appropriately.

- What are the levels of CD9 in cluster 1 and 5?

The plot was corrected to plot Cd9 expression in cluster 1 and 5 as well.

- How are these markers chosen and how do they relate to published isolation strategy published by other single cell studies?

These surface markers were selected based on differentially expressed genes between each of the 5 clusters in our study. Our intent was to find potentially novel markers to isolate these subpopulations. CD55 and CD9 have both been used in Schwalie et al., 2018 and in Hepler et al., 2018 to isolate adipose progenitor populations. However, the combination of CD55, CD9, and CD81 has never been employed to define/isolate an APC subpopulation.

5) Assaying of differentiation potential between sorted cells. The authors have demonstrated nicely that cluster 1, 5, and 6 proliferate at a faster rate than cluster 2 and 4. The differentiation ability was performed by averaging lipid area over 'green objects' which indicate cell number. Wouldn't this approach make clusters 1, 5, 6 have smaller differentiation potential simply because there are more cells in the well? This makes up a major claim of the current study and more evidence demonstrating clusters 2 and 4 indeed differentiate better is required (e.g. QPCRs analysis of differentiation markers between clusters, and images demonstrating the different differentiation potential between clusters. Would the image in Fig 4B suggest that majority of the sorted APCs did not differentiate?). In addition, if the data were acquired using IncuCyte, were there images taken at multiple time points? It will make a stronger argument to present data from different time points (if there are data acquired already).

Indeed, without normalization to cell number (green objects), clusters 2 and 4 have significantly higher differentiation capacities. This is clearly seen in our longitudinal supplementary movies (Supplementary Movies 1-5) that show that BODIPY staining more rapidly accumulates in clusters 2 and 4. Given these cluster growth differences, we maintain that normalization to the number of green objects (nuclei) is necessary because cell number/density is such an important variable affecting adipogenic differentiation.

Minor points

1) The title emphasises on obesity induced alteration to adipose progenitor cell distribution, but most the study focuses on the populations/clusters obtained from normal diet. The last section about DIO results in APC subpopulation shifts etc needs clarification.

We have modified the text and title accordingly.

2) Fig 5 and the relating text are really difficult to follow. In addition, obesity induced changes in

adipose tissue are well known, but there is no discussion of the data obtained in the current study with published work (i.e. what is new and for broad biological significance?)

We have modified the text accordingly in order to improve clarity.

3) More explanations need to be given behind the bioinformatics analysis.

The methods section was modified and bioinformatics analyses are now described in more detail.

4) There is no indication of the number of biological replicates performed in some experiments (eg. QPCRs).

We have added this information in the text/figure legends as requested.

5) Fig 3C-F, do the results indicate that most of the comparisons of gene expression between clusters are not significant? (The error bars are large)

We acknowledge that the error bars of this experiment were relatively large compared to other experiments in the present study. We suspected that it is because this experiment was performed completely independently between each replicate and involves a lot of procedures including tissue/cell preparation, antibody staining, and FACS sorting, that can introduce a lot of errors between biological replicates. Nevertheless, the majority of the comparisons in this gene expression study are significantly different. Also, please note that the highlighted bars indicate significantly different expression in comparison to all other clusters and that this denotation is separate from pairwise significance comparisons indicated by brackets.

6) Fig 2G is missing

Apologies. We have corrected this error.

7) Explanation of the result in Supp Fig 5G is not clear.

We have modified the text accordingly to clarify this result.

8) Should give clear definitions of ASC, PA, and APCs that are used in this study.

The text has been modified to clarify the use of these specific definitions.

9) This study suggested that clusters 1 and 5 are more 'primitive' than clusters 2, 4, and 6 and hence referred clusters 1 and 5 as ASCs. Do the chondrogenic and osteogenic differentiation abilities differ between clusters 1/5 and clusters 2/4/6? (It is not necessary to do these experiment, a discussion is sufficient)

We agree that these are potentially very interesting studies to perform. We have not done these experiments to determine chondrogenic or osteogenic differentiation capacities for our isolated clusters, and we now acknowledge this point in the text.

10) Could the authors comment on the location of these clusters in adipose tissue? (especially clusters 1/5)

11) Do cluster 1/5 and clusters 2/4/6 possess hierarchical relationships? (i.e. do clusters 1/5 give rise to clusters 2/4/6?) It is not essential to demonstrate this by experiments as this is not trivial, discussions would be sufficient.

We appreciate the comment, and we believe that it will be important and interesting to show the relationships between the identified APC subpopulations. Although we have not done the experiments to show this, we now discuss this point in the text along with minor point 10.

November 6, 2019

RE: Life Science Alliance Manuscript #LSA-2019-00561-TR

Dr. Jason Doles
Mayo Clinic
Guggenheim 16-11A1
200 First Street SW
Rochester, MN 55905

Dear Dr. Doles,

Thank you for submitting your revised manuscript entitled "Refining the adipose progenitor cell landscape in healthy and obese visceral adipose tissue".

As you know, I involved original reviewer #1 on your revised version. The reviewer is pleased with most of the changes introduced, but thinks that the clustering method used may not be appropriate and that an alternative method should get tested. This seems feasible in a minor revision, and I would thus like to invite you to further revise your work. When uploading the final version of your manuscript, please also:

- upload the supplementary figures as individual files; the suppl figure legends can get moved into the main manuscript text, please
- provide a short legend for each S T able
- provide a short legend for each movie

A. FINAL FILES:

-- Summary blurb (enter in submission system): A short text summarizing in a single sentence the study (max. 200 characters including spaces). This text is used in conjunction with the titles of papers, hence should be informative and complementary to the title. It should describe the context

and significance of the findings for a general readership; it should be written in the present tense and refer to the work in the third person. Author names should not be mentioned.

B. MANUSCRIPT ORGANIZATION AND FORMATTING:

Sincerely,

Andrea Leibfried, PhD
Executive Editor
Life Science Alliance
Meyerohofstr. 1
69117 Heidelberg, Germany
t +49 6221 8891 502
e a.leibfried@life-science-alliance.org
www.life-science-alliance.org

Reviewer #1 (Comments to the Authors (Required)):

The authors have addressed most of the listed concerns but the performed cell clustering remains not entirely convincing. Specifically, the authors have now generated a within cluster sum of square plot stating that it indicates that 7 is the appropriate cluster number. However, the plot does not show any knee point at all. Furthermore, upon integration of their data with four scRNA-seq datasets (Burl, Hepler, Merrick and Schwalie), we can now see that the subpopulations found by these studies do form related clusters (at least two populations are always consistent (top and bottom)). In contrast, the authors' clustering results do not seem consistent (and each of their clusters is a bit all over the place) (note also though that the authors' selection of colors (i.e. minor differences in tones of green) render it difficult to truly appreciate the complexity of the data). Nevertheless, it still seems that clusters 1 and 5 correspond to the "top" population on the merged tSNE and clusters 2,4,6 to the "bottom" population, but pushing the clustering further, as the authors want to do, seems to not really represent the heterogeneity of the data. It is therefore suggested that the authors could still try other clustering methods that are based on different assumptions than those linked to kmeans clustering, which might ultimately not suit their data.

Reviewer #1 (Comments to the Authors (Required)):

The authors have addressed most of the listed concerns but the performed cell clustering remains not entirely convincing. Specifically, the authors have now generated a within cluster sum of square plot stating that it indicates that 7 is the appropriate cluster number. However, the plot does not show any knee point at all. Furthermore, upon integration of their data with four scRNA-seq datasets (Burl, Hepler, Merrick and Schwalie), we can now see that the subpopulations found by these studies do form related clusters (at least two populations are always consistent (top and bottom)). In contrast, the authors' clustering results do not seem consistent (and each of their clusters is a bit all over the place) (note also though that the authors' selection of colors (i.e. minor differences in tones of green) render it difficult to truly appreciate the complexity of the data). Nevertheless, it still seems that clusters 1 and 5 correspond to the "top" population on the merged tSNE and clusters 2,4,6 to the "bottom" population, but pushing the clustering further, as the authors want to do, seems to not really represent the heterogeneity of the data. It is therefore suggested that the authors could still try other clustering methods that are based on different assumptions than those linked to kmeans clustering, which might ultimately not suit their data.

We thank the reviewer for their helpful comments and overall positive assessment of our revisions. We apologize for the poorly articulated rationale underlying the total within cluster sum of square analysis. Hence, after careful consideration of other methods based on different assumptions, we re-calculated a different coefficient in our clustering, the average silhouette width (Fig S2B). Here, the optimal number of clusters is determined by the highest average silhouette width (Rousseeuw, 1987). This analysis identified $k=2$ as the optimal number of k (clusters) in our data – a choice consistent with prior studies suggesting that adipose progenitor cells are largely composed of two main subpopulations: adipose tissue-derived stem cells (ASCs) and preadipocytes (PAs). Keeping the assumption that our populations were composed of these two main subpopulations, we nevertheless continued downstream analyses with $k=6$ for following reasons: 1) we wanted to select a k value that separated clear outlier cells appearing in our tSNE projections (cells marked with arrows in Fig S2C) into a separate cluster; 2) these outlier cells were clustered in a separate cluster with k greater or equal to 6 (Fig S2A); 3) we wanted to select the minimal k value necessary to achieve this separation so as not to unnecessarily split similar subpopulations; 4) we desired our ideal k to contain at least 3 cells in any cluster to be able to define the cluster with statistical/differentially expressed gene analyses; 5) using $k=7$ (or higher), clusters exist that contain only two cells (Fig S2A) or do not appear in both samples. In summary, the k that best satisfies all these conditions was $k=6$. We clarified this in our main text and made sure to point out that the clearest separation is seen using $k=2$, but that this study aimed to test (using expression analyses AND functional assays) whether or not additional subpopulations exist.

With respect to the scRNA-seq meta-analysis, we apologize for difficult-to-interpret color selection in our tSNE projection in Fig 2E. We changed the coloring scheme to more clearly show where each of our identified clusters appears in the combined tSNE projection. We believe that this should better show that our ASC (cluster 1/5) and PA (cluster 2/4/6) are clearly

separated into top and bottom in the tSNE projection and correspond well with other progenitor subpopulations in published studies (Burl, Hepler, Merrick, and Schwalie).

References

Rousseeuw, P.J. (1987). Silhouettes - a Graphical Aid to the Interpretation and Validation of Cluster-Analysis. *J Comput Appl Math* 20, 53-65.

November 14, 2019

RE: Life Science Alliance Manuscript #LSA-2019-00561-TRR

Dr. Jason Doles
Mayo Clinic
Guggenheim 16-11A1
200 First Street SW
Rochester, MN 55905

Dear Dr. Doles,

Thank you for submitting your Research Article entitled "Refining the adipose progenitor cell landscape in healthy and obese visceral adipose tissue". I appreciate the introduced changes and that an alternative clustering method now employed is consistent with prior studies suggesting that adipose progenitor cells are largely composed of two main subpopulations. It is thus a pleasure to let you know that your manuscript is now accepted for publication in Life Science Alliance. Congratulations on this interesting work.

DISTRIBUTION OF MATERIALS:

Again, congratulations on a very nice paper. I hope you found the review process to be constructive and are pleased with how the manuscript was handled editorially. We look forward to future exciting

submissions from your lab.

Sincerely,
